# Distinct fate, dynamics and niches of renal macrophages of bone marrow or embryonic origins

Fengming Liu [1,2,3 ✉], Shen Dai[3], Dechun Feng[4], Zhongnan Qin[1,2], Xiao Peng[3], Siva S. V. P. Sakamuri[5], Mi Ren[1,2], Li Huang[3], Min Cheng[3], Kabir E. Mohammad[1,2], Ping Qu[3], Yong Chen[3], Chunling Zhao[3], Faliang Zhu[3], Shujian Liang[3], Bertal H. Aktas [6], Xiaofeng Yang[7], Hong Wang[7], Prasad V. G. Katakam [5], David W. Busija[5], Tracy Fischer [1], Prasun K. Datta[3,8], Jay Rappaport[1], Bin Gao[4] & Xuebin Qin [1,2,3 ✉]

Renal macrophages (RMs) participate in tissue homeostasis, inflammation and repair. RMs consist of embryo-derived (EMRMs) and bone marrow-derived RMs (BMRMs), but the fate, dynamics, replenishment, functions and metabolic states of these two RM populations remain unclear. Here we investigate and characterize RMs at different ages by conditionally labeling and ablating RMs populations in several transgenic lines. We find that RMs expand and mature in parallel with renal growth after birth, and are mainly derived from fetal liver monocytes before birth, but self-maintain through adulthood with contribution from peripheral monocytes. Moreover, after the RMs niche is emptied, peripheral monocytes rapidly differentiate into BMRMs, with the CX3CR1/CX3CL1 signaling axis being essential for the maintenance and regeneration of both EMRMs and BMRMs. Lastly, we show that EMRMs have a higher capacity for scavenging immune complex, and are more sensitive to immune challenge than BMRMs, with this difference associated with their distinct glycolytic capacities.

[1] Division of Comparative Pathology, Tulane National Primate Research Center, Covington, LA 70433, USA. [2] Department of Microbiology and Immunology, Tulane University School of Medicine, New Orleans, LA 70112, USA. [3] Department of Neuroscience, Temple University Lewis Katz School of Medicine, Philadelphia, PA, USA. [4] Laboratory of Liver Diseases, National Institute on Alcohol Abuse and Alcoholism, National Institutes of Health, Bethesda, MD, USA. [5] Department of Pharmacology, Tulane University School of Medicine, 1430 Tulane Avenue, New Orleans, LA 70112, USA. [6] Department of Medicine, Brigham and Women's Hospital and Harvard Medical School, Boston, MA, USA. [7] Center for Metabolic Disease Research and Cardiovascular Research, Temple University Lewis Katz School of Medicine, Philadelphia, PA, USA. [8] Present address: Division of Comparative Pathology, Tulane National Primate Research Center, Covington, LA 70433, USA. ✉email: fliu6@tulane.edu; xqin2@tulane.edu

Renal macrophages (RMs) are myeloid cells residing in renal tissue that fulfill specific renal functions including homeostasis, immune surveillance, and repair[1,2]. RMs account for about 50% of total CD45+ leukocytes in mouse kidney and are also found in large numbers in human kidney[3]. They consist of embryo-derived (EMRMs) and bone marrow-derived RMs (BMRMs)[1], but their fate, dynamics, and replenishment have not been systematically investigated[1,2]. Whether these two populations have different functions and metabolic states also remains unclear[1,2]. Addressing these questions will help us to better understand the normal- and patho-biological functions of resident tissue macrophages (TMs), in general, and RMs, in particular.

The paradigm that TMs are differentiated from and continually replaced by bone marrow (BM)-derived monocytes (BMMOs) has been convincingly challenged. Advances in lineage tracing and fate mapping methods have demonstrated that TMs are derived from embryonic progenitors and are seeded before initiation of definitive hematopoietic stem cells[4–6]. Additionally, most TMs maintain their population by in situ proliferation in adulthood and are heterogeneous in different tissues. While it is largely accepted that TMs have three major origins: yolk sac, fetal liver, and BM[4–6], their origin and dynamics can vary greatly from one tissue to another. For example, microglia arise exclusively from yolk sac and are maintained through self-renewal with minimal contribution from BM[7,8]. In contrast, intestinal macrophages derive from fetal liver, rapidly replaced by BMMOs after birth, and continually maintained by BMMOs in adulthood[9].

The origin of TMs may influence their phenotype or functional profile, even within the same organ. Ontogenically distinct TMs also possess different features and functions, especially in diseases. For example, in a murine pancreatic ductal adenocarcinoma model, embryo-derived tumor-associated macrophages exhibit unique pro-fibrotic activities and promoted PDAC progression more vigorously than monocyte-derived tumor-associated macrophages[10]. Another study in colon cancer showed significantly greater expansion of embryo-derived macrophages, as compared to those derived from BM, during tumor progression, suggesting those cells may support tumor development[11]. Understanding the origin and dynamics of TMs under physiological- and pathological conditions will help us to design therapeutic approaches and identify novel targets in macrophage-mediated diseases.

The precise origin, fate, and dynamics of RMs have not been reported. Most TMs derive from embryo and are maintained in a tissue-dependent manner by circulating monocytes, with no or persistent recruitment at a different pace and extent[12]. However, the origin and turnover of TMs are highly tissue-specific and needs to be assessed in each tissue/organ[13]. Further, we have little information on how embryo-derived and BM-derived macrophages contribute to the RM pool[1,2]. How and to what extent BM-derived monocytes contribute to the replenishment of RM populations is also unclear[1]. Experiments using fate mapping and parabiosis have shown that adult RMs are almost exclusively derived from fetal liver monocytes with minimal contribution from peripheral monocytes[14,15]. However, conditional depletion of monocytes by c-myb knockout showed that nearly half of RMs in adult mice are derived from BM within the three months following depletion[6]. This discrepancy may be due to the use of different models, which are not specific to RMs. Additionally, emerging evidence indicates that the macrophages niche in tissues such as brain, liver, and lung may determine the specific functions of the TMs[16,17]. Only a small number of the niche signals have been described for these TMs[16], and the specific molecular mechanism underlying RMs' regeneration has not been studied. Understanding the molecular mechanisms of the RMs niche will

be critical to the development of therapeutics for kidney disease that block or induce specific signaling pathways[16].

The common strategies currently used to manipulate or empty the niche for studying niche signals are irradiation[4], clodronate[18], or diphtheria toxin/diphtheria toxin receptor (DT/DTR)[19,20]. These approaches induce slow cell death (days) by initiating DNA damages and apoptosis[21], which makes it difficult to clearly differentiate cell death from cell regeneration. More importantly, they lack the ability to specifically target RMs of different origins. These limitations preclude emptying specific RMs niches and tracing the origin of regenerated RMs, a requirement for determining the mechanism underlying regeneration of TMs.

Whether the origin of TMs, particularly RMs, dictates their functions and metabolic status remains to be determined[1,16,22]. Kidneys filter blood to remove waste, including immune complex (IC), control the body's fluid balance, and maintain physiological levels of electrolytes. They are also among the first organs to be affected by the disease, such as sepsis[23]. Because RMs are the largest population of immune cells in the kidney, they are critical for detecting and scavenging circulating IC[3] and the host defense against the most common human pathogens[23,24]. Recent evidence indicates that the metabolic status of TMs determines the types of immune response[25,26], but whether this is true for RMs remains unclear.

Here, we report the application of a recently generated Cre-induced-hCD59 transgenic line (ihCD59)[27–29] to trace RMs lineage and determine the fate, dynamics, function, and metabolic states of RMs of BM or embryonic origins. We also couple ihCD59 with rapid ablation of the RMs pool by intermedilysin (ILY) to explore the molecular mechanism underlying RMs niche.

## Results

**Postnatal expansion and maturation of renal macrophages**. To understand the physiological properties of RMs, we characterized RMs by determining their percentage among CD45+ immune cells, total numbers, density (counts/renal weight), proliferative status, and maturation at days 1, 7, 14, 21, 56, and 105 after birth (P1, P7, P14, P21, P56, and P105) in C57BL/B6 mice. We used a standard single-cell flow cytometry method[1,3,30,31] to identify RMs by gating the CD45+CD11b+F4/80high cells (Supplementary Fig. 1a). As illustrated in Fig. 1a, the percentage of RMs (CD45+CD11b+F4/80high) among CD45+ immune cells gradually increased from 28.7% at P1 to 47.2% at P21 and remained at this level up to P105, while the proportion of other myeloid cells (CD11b+F4/80low and CD11b+F4/80−) gradually declined over time. The absolute RMs counts gradually increased from P1 to P21 (neonate and childhood, a stage of gradual body growth), rapidly expanded from P21 to P56 (young adult, a stage of rapid body growth), and remained relatively stable after P56 (adulthood, a stage of slow or no body growth) (Fig. 1b). The densities measured by counts/renal weight did not change significantly from P1 to P56 and increased only slightly between P56 and P105 (Fig. 1c). We next investigated the proliferative status of RMs over time by staining for the proliferative marker, Ki67[9]. We found that RMs from neonate (P1) and juveniles (P21) had a higher percentage of Ki67+ stained cells than RMs from adults (P56 and P91), which shows that RMs proliferate rapidly in early life (Fig. 1d and Supplementary Fig. 1b). These RMs gradually gain expression of MHCII, indicating a postnatal period of maturation[32] (Fig. 1e). Taken together, these data suggest that RMs, as a structural unit of the kidney, expand and mature along with renal growth after birth.

**Origin, fate, and dynamics of renal macrophages**. We used a standard lineage tracing method[33] to investigate the origin of RMs. CX3C chemokine receptor 1 (CX3CR1) is exclusively

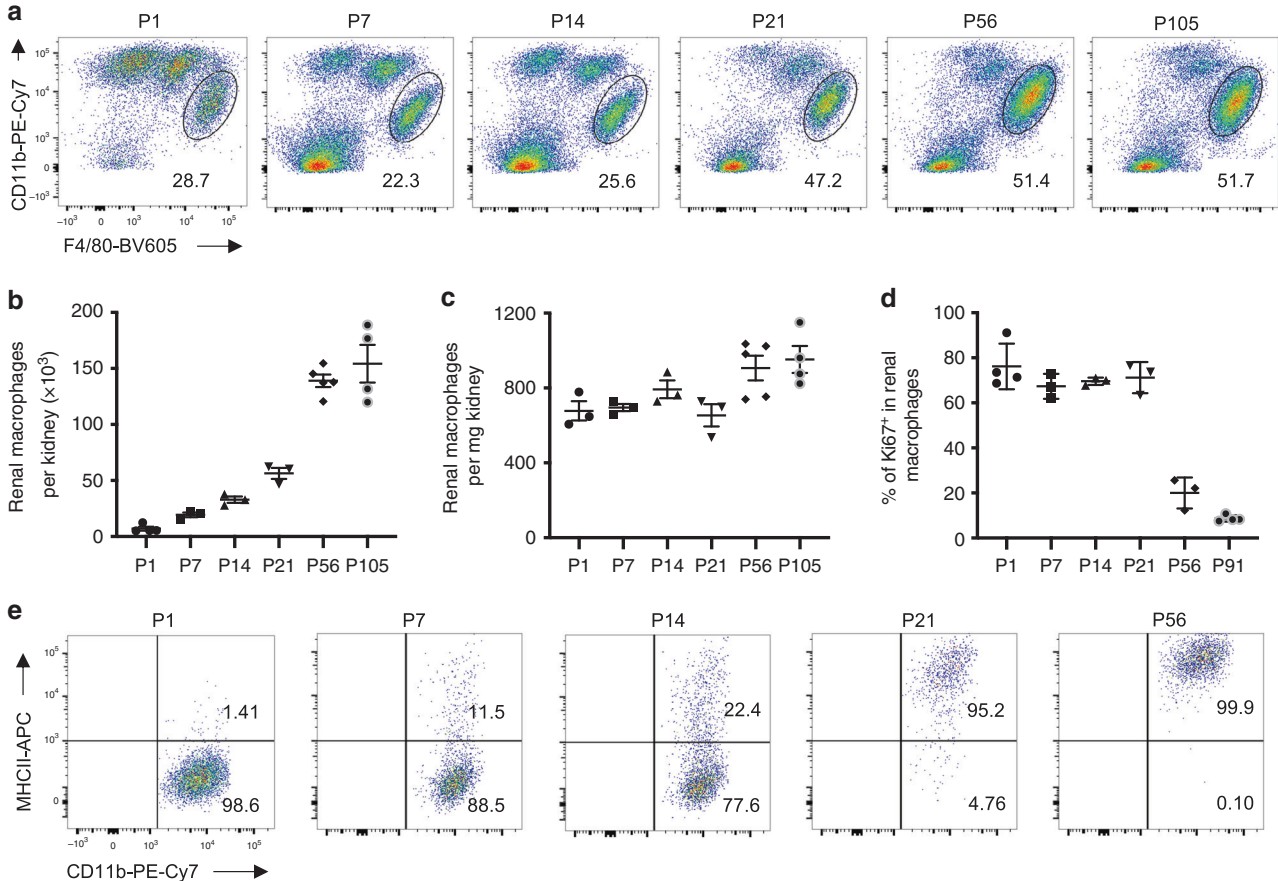

**Fig. 1 Postnatal expansion and maturation of RMs parallel renal growth. a** Representative flow cytometry plots show the percentage of CD11b+F4/80high (cycled area) RMs among CD45 + leukocytes at different ages. **b**, **c** Absolute CD11b+F4/80high RMs cell counts (**b**) and density of RMs (counts/kidney weight) (**c**) at different ages. Data are pooled from two independent experiments (**b**: $n = 3$ mice for P7,14,21; $n = 4$ mice for P1,105; $n = 5$ mice for P56. **c**: $n = 3$ mice for P1,7,14,21; $n = 4$ mice for P105; $n = 5$ mice for P56). **d** Intracellular staining of Ki67 (proliferative marker) in RMs. Ki67 was detected after surface staining of RMs and the percentage of Ki67+ RMs at different ages is quantified. Data are pooled from two independent experiments ($n = 4$ mice for P1,91; $n = 3$ mice for P7,14,21,56). **e** Representative flow cytometry plots show major histocompatibility II (MHCII) and CD11b expression on RMs over time at P1, 7, 14, 21, and 56. All data are presented as mean ± s.e.m. Source data are provided as a Source Data file.

expressed by TMs in kidney, brain, and intestines[19,34–37]. CX3CR1+ macrophages form a contiguous network throughout the cortex and medulla of the kidney and play a critical role in immune surveillance and tissue homeostasis[3,38,39]. CX3CR1 has become a widely accepted marker for labeling RMs[19,37,40]. We utilized *Cx3cr1CreER* mice expressing a Cre-ERT2 fusion protein and enhanced yellow fluorescent protein (EYFP) under the control of endogenous *Cx3cr1* promoter/enhancer elements[36]. Since the EYFP is exclusively observed in CX3CR1-expressing RMs and microglia, it is an in vivo reporter for CX3CR1 expression in mice[19,37]. We also demonstrated that CX3CR1 or EYFP is exclusively expressed in CD45+CD11b+F4/80high RMs of C57BL/B6 or *Cx3cr1CreER+/−* mice, respectively (Supplementary Fig. 1c, d). This result further supports the feasibility and reliability of our approach and model.

To investigate the origin and fate of EMRMs and the changes in the contribution of BM-derived monocytes to the RMs pool with age, we again utilized our previously generated *ihCD59* mice[27]. Human CD59 (hCD59) is a glycophospha-tidylinositol (GPI)-anchored membrane complement regulator that can be stained on the cell surface by anti-hCD59 antibody for flow cytometry[27,41]. We reported previously that *ihCD59* mice express hCD59 only upon Cre-mediated recombination[27]. This allows us to perform a lineage-tracing study of RMs similar to that in Tdtomato (Tdt) fluorescence reporter mice, which is a widely

used lineage-tracing model[42]. To validate this approach, we generated *Cx3cr1CreER+/−/ihCD59+/−* or *Cx3cr1CreER+/−/Rosa26tdtomato+/−* by crossing *Cx3cr1CreER* with either *ihCD59* or *Rosa26-Tdtomato* reporter mice (R26Tdt) to express hCD59 or the reporter Tdt in RMs, respectively. To trace the origin of RMs in the embryo, we induced hCD59 or Tdt expression in RMs of *Cx3cr1CreER+/−/ihCD59+/−* or *Cx3cr1CreER+/−/R26Tdt+/−* fetuses at E8.5, 13.5, and 18.5, respectively. At each of these times, after breeding the *Cx3cr1CreER+/+* females with *ihCD59+/+* or *R26Tdt+/+* males, pregnant females received a single dose of tamoxifen to conditionally induce hCD59 or Tdt expression in all fetuses. The percentage of RMs and microglia conditionally labeled by hCD59 or Tdt at E8.5, 13.5, or 18.5 were traced and analyzed at birth (P0). The percentage of microglia labeled with hCD59 at E8.5, 13.5, and 18.5 was 77.2 ± 0.7%, 96.5 ± 0.8%, and 98.1 ± 0.4% (Fig. 2b, right panel), which were consistent with those using Tdt as a reporter (Fig. 2b, left panel). These results support our notion that *ihCD59* mice are useful for lineage tracing and confirm previous findings that the major origin of microglia is from the embryonic yolk sac[7,15]. The percentage of EMRMs labeled by hCD59 at E8.5, 13.5, and 18.5 was 7.27 ± 0.4%, 46.7 ± 2.5%, and 98.8 ± 0.3% (Fig. 2c). This reveals that the majority of EMRMs appear in the embryos by E13.5 (fetal liver stage) and their genesis is complete by E18.5[4], which indicates that EMRMs derive from fetal liver monocytes.

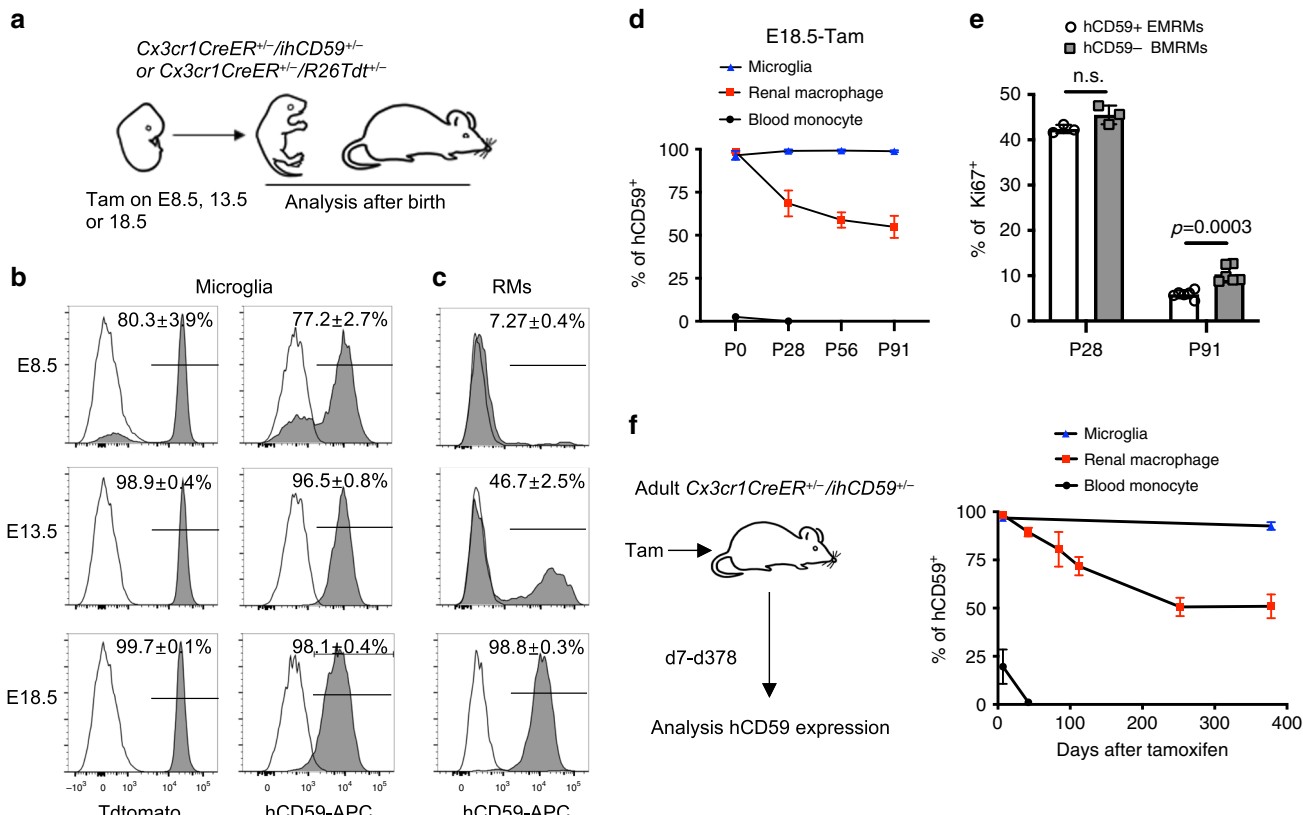

**Fig. 2 RMs are mainly derived from fetal liver and self-maintain through adulthood with a consistent contribution from peripheral monocytes.**
**a** Schematic overview of tamoxifen (Tam) injection at embryonic (E) stage followed by analysis after birth. **b** Histogram of Tdtomato or hCD59 expression by microglia of newborn (P0) *Cx3cr1CreER+/−/R26Tdt+/−* (left) or *Cx3cr1CreER+/−/ihCD59+/−* mice (right) treated with Tam on E8.5, E13.5, or E18.5. Data are pooled from three independent Tam injections (*n* = 5 P0 pups for E18.5-hCD59 group; *n* = 3 P0 pups for all other groups). **c** Histogram of hCD59 expression by RMs of newborn (P0) *Cx3cr1CreER+/−/ihCD59+/−* mice treated with Tam on E8.5, E13.5, or E18.5. Data are pooled from two independent Tam injections (*n* = 3 P0 pups for E8.5/13.5 group; *n* = 5 P0 pups for E18.5 group). **d** The percentages of hCD59+ microglia, renal macrophage and blood monocyte at various time points after labeling on E18.5 (*n* = 5 mice per time point per group). **e** Ki67 staining of hCD59+ embryo-derived renal macrophages (EMRMs) and hCD59- bone-marrow-derived renal macrophages (BMRMs) at P28 and P91 after labeling on E18.5 (*n* = 3 mice for P28; *n* = 6 mice for P91). **f** 2-month-old *Cx3cr1CreER+/−/ihCD59+/−* mice were treated with Tam and analyzed for hCD59+ microglia, renal macrophage, and blood monocyte from 7 to 378 days after labeling. The percentages of hCD59+ microglia, renal macrophage, and blood monocyte are quantified. Data are pooled from two independent experiments (*n* = 4 mice per time point per group). All data are presented as mean ± s.e.m. *p*-values by two-tailed unpaired *t*-test are indicated in **e**, n.s. indicates *p* > 0.05. Source data are provided as a Source Data file.

We investigated the fate of EMRMs in further detail and explored how BM-derived monocytes contribute to the pool of RMs at different developmental stages. To study the fate of RMs from birth to full maturity (P0 to P91), we used hCD59 to label the EMRMs of *Cx3cr1CreER+/−/ihCD59+/−* at E18.5 after a single tamoxifen injection to pregnant *Cx3cr1CreER+/+* females bred with *ihCD59+/+* males. After birth, we periodically (P0, P28, P56, and P91) monitored the hCD59+ and hCD59− resident macrophages in kidney (RMs) and brain (microglia) (Fig. 2d). Importantly, BMRMs are hCD59- in these tissues since they have never been exposed to tamoxifen. In agreement with a previous report[36], hCD59+ circulating monocytes were completely replaced by newly generated hCD59- monocytes 4 weeks after tamoxifen exposure, whereas 100% of microglia remained hCD59+ from P1 to P91. These results further support the current view that BM monocytes makes little to no contribution to microglia pool[8]. The percentage of hCD59+ EMRMs gradually decreased to 60% at P28 and maintained at stable levels (~50–60%) at least until P91 (Fig. 2d).

To trace the fate of RMs with aging (from P56 to P378), we used hCD59 to label RMs after three tamoxifen injections to 56-day-old *Cx3cr1CreER+/−/ihCD59+/−* mice and analyzed hCD59 expression

in CX3CR1+ cells in multiple tissues, including brain, blood, spleen, lung, kidney, BM, heart, liver, and peritoneum at 2 or 30 days after tamoxifen injection (Supplementary Fig. 2). The percentage of hCD59+ RMs was stably maintained at day 30 after tamoxifen injection, which is comparable to the finding that 100% of microglia are hCD59+ (Supplementary Fig. 2). These results confirm previous findings that a high percentage of RMs are long-lived[43]. To better understand the dynamics of EMRMs and BMRMs with age, we sequentially traced and analyzed hCD59+ and hCD59- RMs and microglia (at day 7, 42, 91, 102, 252, and 378 post labeling). As expected, we observed that 100% of microglia were hCD59+ even at P378 and that 100% of monocytes became hCD59- at P42 (Fig. 2f, right panel). Interestingly, we also found that the percentage of hCD59+ RMs gradually decreased to ~60% from P7 to P252 and was maintained at this stable level until P378 (Fig. 2f, right panel), which is consistent with the finding in *Cx3cr1CreER+/−/ihCD59+/−* induced by tamoxifen at E18.5 (Fig. 2d). These results were also confirmed by tracing and analyzing reporter Tdt in the RMs of *Cx3cr1CreER+/−/R26Tdt+/−* after tamoxifen injection (Supplementary Fig. 3a). Anatomically, immunofluorescent (IF) staining of Cx3Cr1-EYFP in kidney sections from *Cx3cr1CreER+/−* mice showed that CX3CR1+

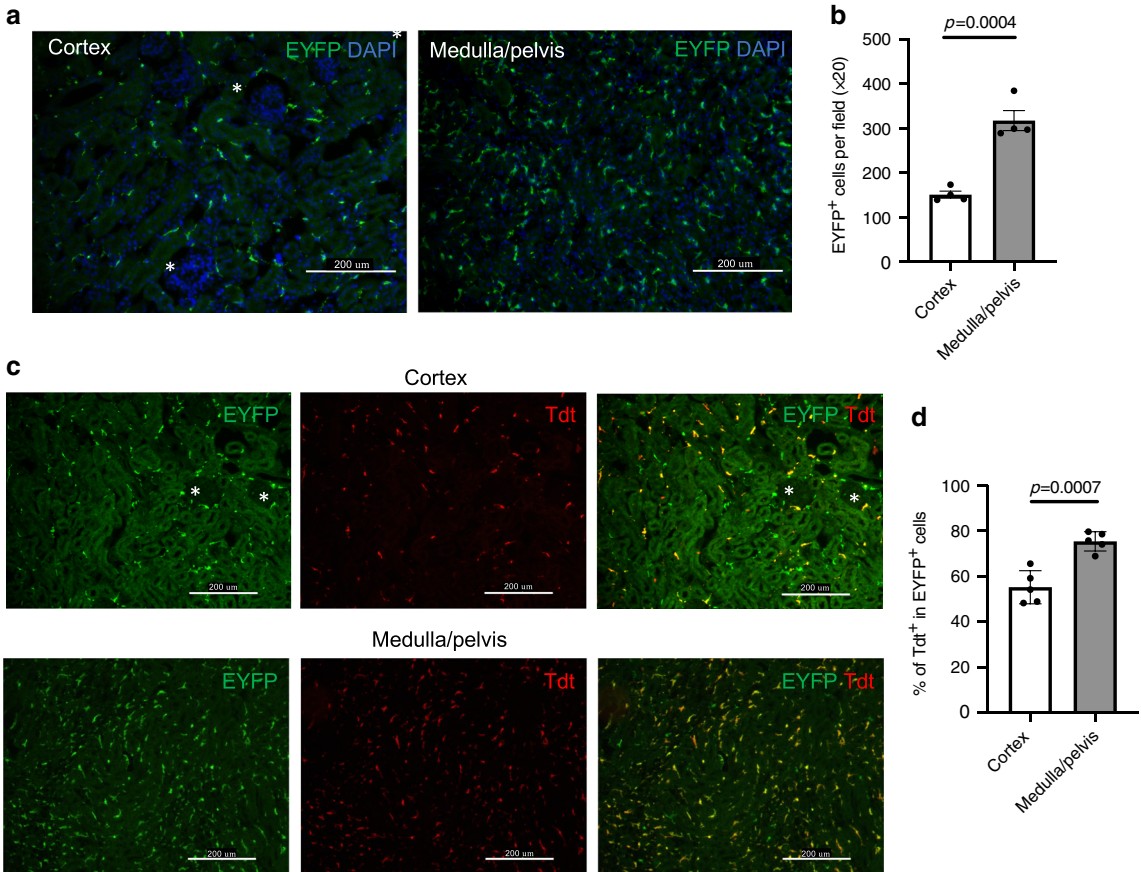

**Fig. 3 Localization of EMRMs and BMRMs in kidney. a** Representative immunofluorescence (IF) staining of EYFP (green) and DAPI (blue) and **b** quantification of EYFP+ cells in kidney cortex and medulla/pelvis of adult *Cx3cr1CreER*$^{+/-}$ mice. Asterisk indicate glomerulus in cortex. Each dot in **b** represents the average cell numbers per field (×20 magnification) from one mouse (mean ± s.e.m., two-tailed unpaired *t*-test, $n = 4$ mice per group). **c** Representative IF staining of EYFP (green) and Tdtomato (Tdt, red) in cortex and medulla/pelvis of adult E18.5-Tam-*Cx3cr1CreER*$^{+/-}$/*R26Tdt*$^{+/-}$ mice. Yellow staining cells in merged image are EMRMs. Asterisk indicate glomerulus in cortex. **d** Quantification of the percentage of Tdt+ EMRMs in all RMs of cortex and medulla/pelvis. Each dot represents the average cell numbers per field (×20 magnification) from one mouse (mean ± s.e.m., two-tailed unpaired *t*-test, $n = 5$ mice per group). Source data are provided as a Source Data file.

RMs spread throughout the cortex and medulla/pelvis, which is consistent with previous study (Supplementary Fig. 4)[38]. Interestingly, we observed that the cell density of EYFP + RMs in medulla/pelvis was significantly higher than that of cortex (Fig. 3a, b). Furthermore, kidney sections from E18.5-Tam-*Cx3cr1CreER*$^{+/-}$/*R26Tdt*$^{+/-}$ revealed that EMRMs is preferentially located in medulla/pelvis, as evidenced by the significant higher percentage of Tdt+ labeling in medulla/pelvis than that in cortex. (Fig. 3c, d). Taken together with the findings obtained from the embryonic labeling and tracing studies, these results demonstrate that the pool of RMs consists of long-lived EMRMs, and BMRMs with a brief influx after birth and a slow contribution at adult stages by peripheral monocytes.

**Origin and longevity of newly regenerated renal macrophages**. As shown in Supplementary Fig. 5a, b, RMs are highly vulnerable to ischemia/reperfusion (I/R) injury, a common complication of pathological conditions such as shock and organ rejection[44]. Therefore, understanding how RMs are replenished after emptying their niche is clinically relevant and important for developing better therapies. To manipulate and empty the macrophage niche in kidney, we took advantage of the unique interaction between ILY and hCD59, which mediates rapid ablation of hCD59-expressing cells in mice[27,41]. ILY, a toxin secreted by *Streptococcus intermedius*, binds exclusively to hCD59[41,45,46] and, after binding, forms toxin

pores that lyse the cells within seconds[27,41,46]. Administration of ILY to *ihCD59*+ crossed with various Cre-driver lines rapidly and specifically ablates immune and epithelial cells without off-target effects[27–29]. The rapidity of the ablation, a unique feature of this model, makes this strategy particularly apt for determining the dynamic regeneration of the targeted cells after ablation[27–29,41]. To specifically and rapidly empty the RMs niche in kidney with minimal off-target effects in other tissues, we chose to inject ILY 30 days after tamoxifen induction in *Cx3cr1CreER*$^{+/-}$/*ihCD59*$^{+/-}$ mice. There were two reasons for this choice: (1) on day 30 after tamoxifen injection, almost 100% of CX3CR1 + cells continue to express hCD59 in brain and kidney of *Cx3cr1CreER*$^{+/-}$/*ihCD59*$^{+/-}$ mice, whereas no hCD59+ cells are present in blood, spleen, lung, BM, and liver, and less than 50% of macrophages are hCD59+ in heart and peritoneum (Supplementary Fig. 2), (2) ILY, a 54-kd protein, cannot pass the blood-brain barrier to ablate hCD59+ microglial cells[27]. As shown in Fig. 4a, b and Supplementary Fig. 3b, RMs were efficiently depleted one day after ILY injection in *Cx3cr1CreER*$^{+/-}$/*ihCD59*$^{+/-}$, but not in control *Cx3cr1CreER*$^{+/-}$/*ihCD59*$^{-/-}$ mice. The population starts to be replenished at day 3 after injection and recovers to 50% of its original level at day 7 (Fig. 4a, b).

We then determined the origin of the regenerated RMs. These may originate from monocytes as seen for Kupffer cells[20]; in situ proliferation of resident cells, as seen for regenerated microglia[19]; or both monocytes and residual cells, as seen after depletion of

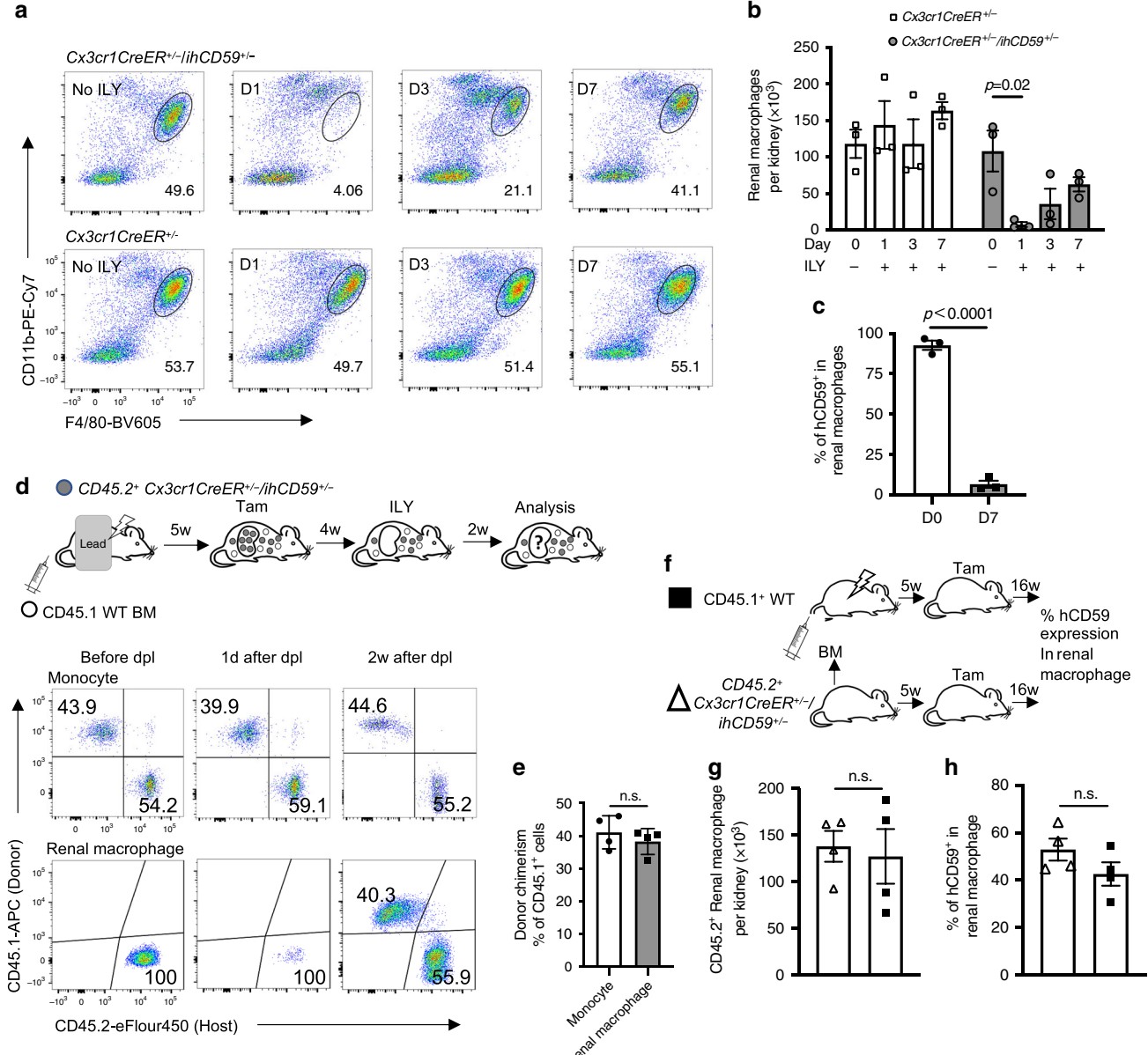

**Fig. 4 Peripheral monocytes rapidly differentiate into long-lived RMs after niche is emptied. a** Representative flow cytometry plots of the percentage of CD11b+F4/80high (cycled area) RMs among CD45+ leukocytes and **b** RMs cell counts at different days (D1, D3, and D7) after ILY injection (120 ng/g body weight, i.v.) or without ILY (No ILY). Data are representative of two independent experiments (*n* = 3 mice per time point per group). **c** hCD59 expression on original (D0) and regenerated RMs (D7 after ILY; *n* = 3 mice per group). **d** Schematic overview of generation and treatment of chimeric mice by partially irradiating recipient (CD45. 2+) male mice followed by reconstitution with the CD45.1+ wild type (WT), bone marrow (BM), and ILY injection. Dot plots show CD45.1 (Donor) and CD45.2 (Host) expression on peripheral monocyte (upper panels) and RMs (lower panels) before, 1 day (d) or 2 weeks (w) after ILY-mediated RMs depletion (dpl). **e** No significant difference in donor chimerism (CD45.1/CD45.1+CD45.2) between monocyte and regenerated RMs at 2 weeks after ILY injection (also shown in the right panel of the dot plots, *n* = 4 mice per group). **f** Schematic overview of bone marrow transplantation (BMT) and tamoxifen (Tam) injection. **g** Cell counts of CD45.2+ RMs in age-matched CD45.2+*Cx3cr1CreER*+/−/*ihCD59*+/− and chimeric CD45.1+ male mice 5 weeks after BMT (*n* = 4 mice per group). **h** hCD59 expression on original CD45.2+ RMs and regenerated RMs in chimeric mice 16 weeks after tamoxifen (*n* = 4 mice per group). All data are presented as mean ± s.e.m. *p*-values by two-tailed unpaired *t*-test are indicated in **b**, **c**, **e**, **g**, **h**, n.s. indicates *p* > 0.05. Source data are provided as a Source Data file.

alveolar macrophages[47]. Only ~5% of newly generated RMs expressed hCD59, indicating a minimal contribution of in situ proliferation to the repopulation (Fig. 4c). To conclusively demonstrate the origin of RMs after the rapid emptying, we created chimeric mice by partially irradiating *Cx3cr1CreER*+/−/ *ihCD59*+/− recipient (CD45.2+) mice, which were reconstituted with CD45.1 + B6 donor BM. To prevent radiation damage of the CD45.2+ RMs, the abdomen of *Cx3cr1CreER*+/−/*ihCD59*+/− was

covered by a lead shield during the irradiation (upper panel in Fig. 4d). This procedure enabled us to develop mice with chimeric peripheral monocytes and intact recipient CD45.2+ RMs, as determined by the ratio of CD45.1+ vs. CD45.2+ cells. We found that after 5 weeks of reconstitution, the pools of the peripheral monocytes or RMs were composed of mixed CD45.1+ and CD45.2+ cells (43.9% and 54.2%) or intact CD45. 2 cells (100% CD45.2), respectively (left panel of the dot plots in

Fig. 4d). To empty the niche, we injected the chimeric $Cx3cr1CreER^{+/-}/ihCD59^{+/-}$ mice with tamoxifen to exclusively label CD45.2+ RMs with hCD59 and injected ILY 4 weeks later. The ILY treatment rapidly emptied the pool of RMs but not peripheral monocytes (middle panel of the dot plots in Fig. 4d). We analyzed and compared the chimerisms of regenerated RMs and peripheral monocytes at 2 weeks post-depletion when most of the RMs have been replaced by newly regenerated RMs (right panel of the dot plots in Fig. 4d). We found that the chimerism of regenerated RMs was similar to that of monocytes (Fig. 4e). This indicates that most regenerated RMs derive from peripheral monocytes and far fewer from local proliferation of CD45.2+ cells. If local proliferation of CD45.2+ cells contributed to the pool of the regenerated RMs, the ratio of CD45.2 cells should have been greater. Together, these findings conclusively demonstrate that peripheral monocytes rapidly differentiate to RMs after their niche in kidney is emptied. Our results also indicate that coupling the $ihCD59^{27}$ with rapid ILY-mediated ablation of the RMs niche is an exquisitely suitable strategy for tracing the lineage of regenerating RMs.

We have shown that RMs are long-lived (Fig. 2d, f), but whether the newly generated BMRMs are as long-lived as the original RMs (EMRMs and BMRMs) is unclear[5]. To address this question, we emptied the kidney niche in CD45.1+B6 mice with a lethal dose of radiation (Supplementary Fig. 5c), then reconstituted it with donor BM cells from CD45.2+ $Cx3cr1CreER^{+/-}/ihCD59^{+/-}$ to generate grafts of tamoxifen-induced labelable RMs in CD45.1 + B6 mouse hosts (regenerated RMs) (upper panel in Fig. 4f). The regeneration was completed at 5 weeks after the BM transplantation (BMT), as evidenced by the observation that the numbers of regenerated CD45.2+ $Cx3cr1CreER^{+/-}/ihCD59^{+/-}$ RMs in the chimeric CD45.1 + B6 hosts were similar to those in donor mice (Fig. 4g). Therefore, we injected the chimeric CD45.1 + B6 mice with tamoxifen to label RMs with hCD59 at 5 weeks after the BMT and traced the stability from day 1 to 16 weeks. The percentage of regenerated hCD59+ RMs in the chimeric mice gradually decreased from 100% to 40%, which is comparable to that of the original hCD59+ RMs in age-matched $Cx3cr1CreER^{+/-}/ihCD59^{+/-}$ mice (50%) (Fig. 4h). In addition, the radiation and aging did not induce significant changes in the glomerular filtration rate (GFR), a sensitive clinical biomarker for kidney function[48], as compared with their respective control mice (Supplementary Fig. 6). These results demonstrate that after emptying the RMs niche in kidney, the newly generated BMRMs are long-lived and preserve stability comparable to that of the original RMs.

**CX3CR1/CX3CL1 signaling is required for RMs regeneration.** Our demonstration that peripheral monocytes are able to migrate and differentiate into long-lived RMs after emptied the RMs niche prompted us to investigate the molecular mechanism underlying RMs regeneration. Proper regeneration and maintenance of RMs niche are critical for kidney homeostasis[1,24]. Monocyte migration depends on chemokines[49]. To identify which chemokines are critical for the migration of monocytes to kidney, we measured 14 chemokines in the serum of $Cx3cr1CreER^{+/-}/ihCD59^{+/-}$ and control $Cx3cr1CreER^{+/-}$ mice 1 day after RMs depletion. Of these, only CX3CL1 was significantly increased as compared to control mice. Serum CX3CL1 level was elevated on days 1, 2, and 3 and gradually declined to pre-depletion levels by day 7 after RMs depletion (Fig. 5a and Supplementary Fig. 7a). Importantly, CX3CL1 levels measured in the kidney were also significantly increased on day 1 after RMs depletion (Fig. 5b). These results suggest that CX3CL1 serves as a niche-signaling molecule that is released from kidney to attract

monocytes after RMs loss. We also observed a consistent increase in serum CX3CL1 after kidney I/R injury (Supplementary Fig. 7b).

CX3CR1 is the only known receptor for CX3CL1 and, like CCR2, plays a major role in directing monocyte migration from blood to tissue[50]. As such, we next investigated the effect of CX3CR1 and CCR2 on RMs development and repopulation. Consistent with previous results[1,24,51], we observed that 56-day-old homozygous $Cx3cr1CreER^{+/+}$, in which $Cx3cr1$ was deficient ($Cx3cr1^{-/-}$), had significantly fewer RMs, but not microglia, than heterozygous $Cx3cr1CreER^{+/-}$ mice ($Cx3cr1^{+/-}$) (Supplementary Fig. 8a, b). The number of RMs in $Cx3cr1^{-/-}$ mice were also reduced at P1, P28, and P98 (Supplementary Fig. 8b). In contrast, $Ccr2^{-/-}$ mice had a similar number of RMs and microglia (Supplementary Fig. 8c). This supports the notion that the CX3CR1/CX3CL1 axis is crucial for embryonic development and persistent maintenance of RMs under steady-state conditions[1] and that CCR2 is dispensable for maintaining the RMs in a steady state. We also emptied the niche with ILY-induced RMs depletion and irradiation to investigate whether CX3CR1 is required for RMs repopulation. For this purpose, we utilized either the homozygous $Cx3cr1CreER^{+/+}$ as CX3CR1-deficient mice ($Cx3cr1^{-/-}$) and heterozygous $Cx3cr1CreER^{+/-}$ mice as CX3CR1-sufficient mice ($Cx3cr1^{+/-}$) in combination with $ihCD59^{+/-}$ ($Cx3cr1CreER^{+/+}/ihCD59^{+/-}$ and $Cx3cr1CreER^{+/-}/ihCD59^{+/-}$). $Cx3cr1CreER$ animals were generated by knocking in CreER to exon 2 of Cx3cr1, which leads to complete CX3CR1 deficiency[36] in homozygous, but not heterozygous mice. The ratio and number of the regenerated RMs were significantly reduced in $Cx3cr1CreER^{+/+}$ but not $Cx3cr1CreER^{+/-}$ mice at 7 days after ILY-mediated RMs depletion (Fig. 5c). Inhibition of CCR2 activity in a similar experiment performed in $Cx3cr1CreER^{+/-}/ihCD59^{+/-}$ mice did not influence the repopulation (Supplementary Fig. 8d). To confirm these results, we depleted RMs by irradiating CD45.1+ recipient mice, followed by reconstituting with CD45.2+ WT, $Cx3cr1^{-/-}$ or $Ccr2^{-/-}$ BM cells, and determined the effect of chemokine receptors deficiency on RMs repopulation (Fig. 5d). Regeneration of RMs and monocytes in kidney, however, was significantly reduced in chimeric mice reconstituted with $Cx3cr1^{-/-}$ BM (Fig. 5e), but not $Ccr2^{-/-}$ BM (Fig. 5f), as compared with chimeric mice reconstituted with WT BM at 5 weeks after grafting (Supplementary Fig. 9). Interestingly, CX3CR1 deficiency did not affect the newly generated TMs and infiltrating monocytes in lung and spleen (Fig. 5e). We also observed a significantly reduced number of infiltrating monocytes in kidney, lung, and liver in mice reconstituted with $Ccr2^{-/-}$ BM, which confirms the previous finding that CCR2 is critical for monocyte infiltration[52]. Taken together, these data demonstrate that the CX3CR1/CX3CL1 axis is a specific niche signaling pathway that is not only required for RMs development and maintenance in steady state but also indispensable for RMs repopulation under pathologic condition.

**Functional analysis of two lineage-derived renal macrophages.** Functional differences between resident EMRMs and BMRMs in the kidney have not been reported[2]. Geissmann and colleagues recently presented evidence that RMs play a role in immune surveillance[3]. They injected mice with freshly prepared anti-BSA IgG:BSA (IC) complex to demonstrate that RMs efficiently take up IC and trigger an inflammatory response, which fulfills their physiological monitoring function[3]. We followed a similar approach to investigate the IC uptake capacities of EMRMs and BMRMs. As described above and shown in Fig. 2, we differentiated EMRMs (hCD59+) from BMRMs (hCD59−) with hCD59 induced by tamoxifen at E18.5 on $Cx3cr1CreER^{+/-}/ihCD59^{+/-}$ mice. As expected from the previous report[3], RMs

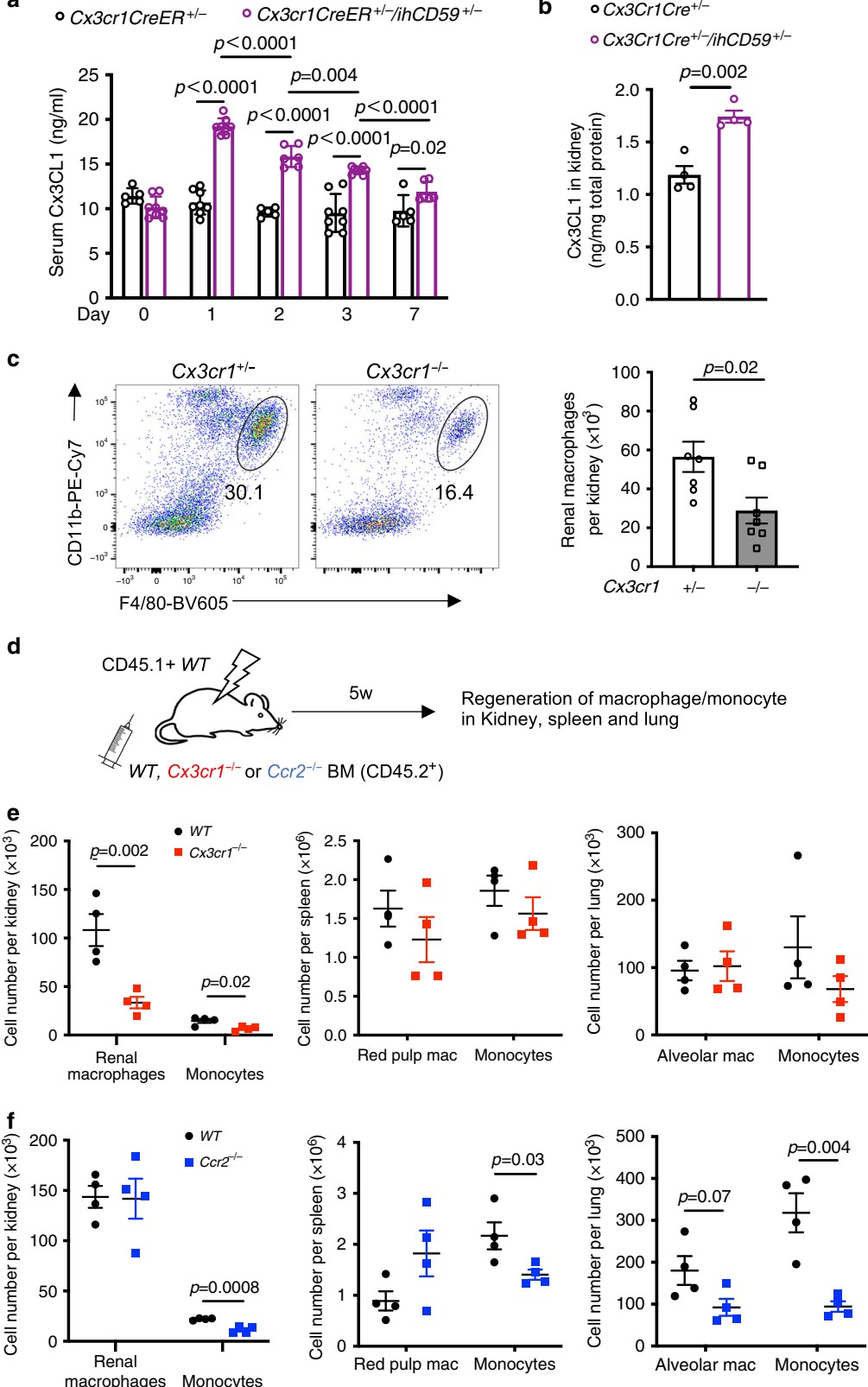

were the only myeloid cell among three major CD11b+ myeloid populations present in kidney responsible for the uptake of Alexa647-IC 2 h after injection (Fig. 6a). Surprisingly, hCD59+ EMRMs had a significantly higher mean fluorescence intensity (MFI) of IC than hCD59- BMRMs (Fig. 6b, c). This was confirmed in *Cx3cr1Cre*[+/-]/R26Tdt[+/-] mice (Fig. 6d), suggesting

that EMRMs carry out immune monitoring more efficiently than BMRMs. RMs Fc receptor density, including FcγRI, FcγRII/III and FcγRIV, cannot explain intrinsic functional capacities since the expression levels of these receptors were not significantly different (Fig. 6e). Staining of macrophage/monocyte markers showed that the expression level of CX3CR1 and F4/80 are higher

**Fig. 5 CX3CR1/CX3CL1 axis is required for RMs regeneration under normal and pathologic conditions. a** Serum CX3CL1 level after ILY injection (120 ng/g b.w.) in $Cx3cr1CreER^{+/-}$ and $Cx3cr1CreER^{+/-}/ihCD59^{+/-}$ mice. Day-0 indicates no ILY injection. Data are pooled from three independent experiments ($Cx3cr1CreER^{+/-}$ group: $n = 5$ mice for Days 0, 2, and 7 and $n = 8$ mice for Days 1 and 3; $Cx3cr1CreER^{+/-}/ihCD59^{+/-}$ group: $n = 8$ mice for Days 0, 1, and 3 and $n = 5$ mice for Days 2 and 7). **b** CX3CL1 level in kidney homogenate at 1-day post ILY injection. CX3CL1 protein level was normalized by total protein of homogenate ($n = 4$ mice per group). **c** Percentage of RMs (circled area) in CD45+ cells (left, dot plots) and absolute cell counts (right, column graph) at 7 days after ILY injection under CX3CR1 sufficient ($+/-$) and deficient ($-/-$) conditions. Data are pooled from two independent experiments. ($n = 7$ mice per group). **d** Schematic overview of the experiment. CD45.1+ WT male mice were lethally irradiated and reconstituted with WT, $Cx3cr1^{-/-}$ or $Ccr2^{-/-}$ BM (CD45.2+). Regeneration of tissue macrophages (mac) or monocytes in kidney, spleen, and lung was determined 5 weeks later. **e, f** Number of donor-derived TMs and monocytes in kidney, spleen, and lung 5 weeks after BMT. **e** WT vs $Cx3cr1^{-/-}$ (**f**) WT vs $Ccr2^{-/-}$ ($n = 4$ mice per group). All data are presented as mean ± s.e.m. p-values by two-tailed unpaired t-test are indicated in **a–c, e, f**. Source data are provided as a Source Data file.

on EMRMs than BMRMs, whereas CCR2 and CD11c are higher on BMRMs than EMRMs (Fig. 6f).

We next investigated the response of EMRMs and BMRMs to immune challenge in a disease setting. TMs act as front line defenders against immune challenge and have been shown to scavenge IC[3], and respond to anti-glomerular basement membrane (GBM) glomerulonephritis[53] and lipopolysaccharide (LPS)-induced acute kidney injury (AKI)[54]. Utilizing CD86 and TNF as indicators of macrophage activation[3], we monitored the response of EMRMs and BMRMs to these challenges in vivo. Consistent with greater uptake of IC (Fig. 6b–d), EMRMs expressed significantly higher levels of CD86 and TNF than BMRMs 2 h after IC injection (Fig. 7a, b). Similarly, in response to anti-GBM induced nephritis, EMRMs had higher CD86 expression and TNF production than BMRMs at 2 and 24 h after anti-GBM serum injection (Fig. 7c, d). This response by RMs occurred in parallel with the onset of disease, as indicated by increased urine albumin-to-creatinine ratio (ACR) at 2 h after anti-GBM serum injection (Supplementary Fig. 10a). Urine ACR is the early evidence of renal dysfunction[55] and happened before the elevation of serum blood urine nitrogen (BUN), which was not observed until day 2 after challenge (Supplementary Fig. 10b). We did not observe a significant change of the localization of RMs before and 24 h after anti-GBM serum injection (Supplementary Fig. 10c, d). In addition, since many autoimmune and kidney diseases have gender differences in prevalence and clinical features[56], we compared the response of RMs to anti-GBM induced nephritis in males and females. We found a delayed disease onset in male mice, indicated by our observation that males had significantly lower urine ACR than females at 2 h after anti-GBM serum injection (Supplementary Fig. 11a). Consistently, males had significantly lower CD86 levels on EMRMs and BMRMs than females at 2 h after challenge (Supplementary Fig. 11b). In LPS-induced AKI model, LPS challenge induced higher level of TNF in EMRMs than that in BMRMs in vivo (Fig. 7e). Together, these results indicate that EMRMs are more sensitive to immune challenge than BMRMs and the activation of RMs is associated with disease progression.

In pathogen-induced AKI, the activation of RMs and subsequent release of inflammatory cytokines can lead to progressive kidney inflammatory injury and fibrosis[23,24]. LPS and lipoteichoic acid (LTA) are major components of the outer membrane of gram-negative and gram-positive bacteria, respectively, and poly (I:C) is an immune-mimic double-stranded RNA virus[57]. We compared the response to these immunogens by EMRMs and BMRMs from E18.5-Tam-$Cx3cr1Cre^{+/-}/ihCD59^{+/-}$ mice by measuring TNF and IL-6, two major inflammatory cytokines that play a dominant role in kidney diseases[58]. EMRMs released significantly higher levels of TNF and IL-6 than BMRMs in response to LPS and LTA ex vivo (Fig. 7f, g). Additionally, EMRMs released significantly higher levels of TNF but not IL-6, than BMRMs in response to poly (I:C) (Fig. 7f, g). Consistently,

EMRMs isolated from E18.5-Tam-$Cx3cr1CreER^{+/-}/R26Tdt^{+/-}$ mice also released higher levels of both TNF and IL-6 in response to LPS ex vivo (Supplementary Fig. 12). Interestingly, even in the absence of immune challenge, EMRMs secreted more TNF than BMRMs (Fig. 7f), while little change in cytokine secretion was observed in BMRMs, regardless of challenge (Fig. 7f, g). This may suggest a greater intrinsic pro-inflammatory status of EMRMs that is more responsive to immune challenge. Together, these findings indicate that EMRMs have a higher capacity for scavenging IC and are more sensitive to immune challenges than BMRMs. To our knowledge, this is the first demonstration of functional and phenotypic differences between RMs from two distinct origins.

**Metabolic analysis of two lineage-derived renal macrophages.** Extensive evidence indicates that specific changes in cellular metabolism are directly associated with macrophage cytokine release in response to pathogens[59] and that the metabolic state of macrophages influences various types of immune responses[25,26]. This motivated us to explore the mechanisms underlying the immune responses of the RMs populations by metabolic analysis of EMRMs vs. BMRMs. We simultaneously monitored the two major energy-producing pathways, glycolysis and mitochondrial respiration, by measuring the change of extracellular acidification rate (ECAR) and oxygen consumption rate (OCR), respectively. ECAR (Fig. 8a) and OCR (Fig. 8b) were significantly higher in EMRMs than BMEMs at baseline and under stress (introduction of oligomycin/FCCP after three baseline measurements), indicating that EMRMs have a higher capacity to produce energy through both glycolysis and mitochondrial respiration.

We further explored the different metabolic profiles of EMRMs and BMRMs under the pathological condition, such as IC challenge. Total RMs from IC-treated mice had more ECAR than RMs from control mice (Supplementary Fig. 13). This also supports previous observation of metabolic reprogramming in activated macrophages challenged with LPS[60]. These results indicate that macrophages can shift energy metabolism from mitochondrial respiration to glycolysis under pathological conditions[60], which makes them resilient to survive low oxygen microenvironments. We consistently observed that EMRMs have higher ECAR than BMRMs after IC stimulation (Fig. 8c). This is congruent with the finding that EMRMs have a higher glucose uptake than BMRMs under static and stimulating conditions ex vivo (Fig. 8d), which ties metabolic activation of EMRMs and BMRMs with their glucose uptake. Finally, we inhibited glycolysis with 2-deoxy-d-glucose (2-DG), an analog of glucose that competitively inhibits production of glucose-6-phosphate, and assessed the effect on TNF release by EMRMs and BMRMs in response to LPS. Both RMs populations produced significantly less TNF when treated with 2-DG, as compared to the absence of 2-DG (Fig. 8e), although EMRMs released higher levels of TNF than BMRMs irrespective of glycolysis inhibition (Fig. 8e). These findings further support the impression that EMRMs are

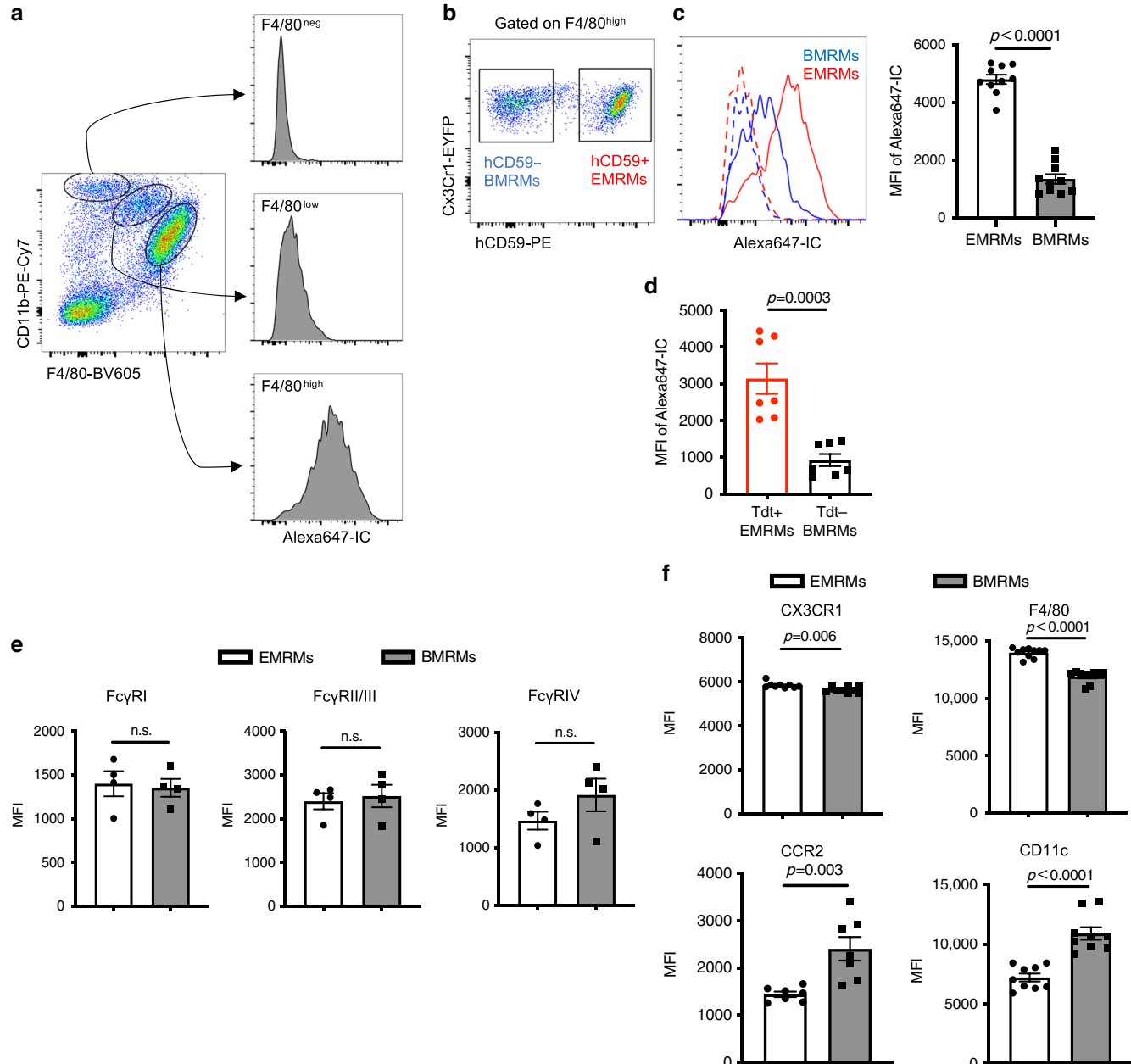

**Fig. 6 EMRMs have higher capacity to scavenge immune complex than BMRMS. a** Flow cytometry analysis of BSA-RαBSA immune complex (IC) uptake by renal myeloid cells at 2 h after i.v. injection of 50 μg BSA-Alexa 647 + 110 μg RαBSA complex. **b** Gating strategy of EMRMs and BMRMs by hCD59 expression. **c** Flow cytometry analysis of BSA-RαBSA (Alexa647-IC) uptake by hCD59+ EMRMs and hCD59- BMRMs 2 h after injection (left: representative histogram. right: mean fluorescent intensity [MFI]). Data are pooled from three independent experiments. ($n = 10$ mice per group). **d** MFI of BSA-RαBSA (Alexa-647-IC) in Tdt+ EMRMs and Tdt- BMRMs 2 h after injection. Data are pooled from two independent experiments ($n = 7$ mice per group). **e** MFI of Fcγreceptors expression on hCD59+ EMRMs and hCD59- BMRMs ($n = 4$ mice per group). **f** MFI of CX3CR1, CCR2, F4/80 and CD11c on hCD59+ EMRMs and hCD59− BMRMs ($n = 10$ mice for F4/80; $n = 9$ mice for CX3CR1 and CD11c; $n = 7$ mice for CCR2). All data are presented as mean ± s.e.m. $p$-values by two-tailed unpaired $t$-test are indicated in **c**, **d**, **e**, **f**, n.s. indicates $p > 0.05$. Source data are provided as a Source Data file.

more sensitive to immune challenge than BMRMs. Moreover, the level of responsiveness is associated with a greater glycolytic capacity, indicating that the glycolytic states of EMRMs and BMRMs contribute significantly to their ability to respond to immune challenge.

## Discussion

Lineage tracing provides a powerful tool for understanding tissue development, homeostasis, and disease, particularly when it is combined with experimental manipulation of the cell populations of interest[3,33,61]. The newly developed lineage-tracing methods have advanced our understanding of the biology of TMs considerably[4–6]. Here, we took advantage of the superiority of ILY/ihCD59-mediated rapid cell ablation[27–29,41] to map the cell fate of RMs and study the origin and maintenance of RMs niches. We demonstrate that inducibly expressed hCD59 is very stable and can be readily visualized by antibody staining. The fate, origin, dynamics and function of RMs revealed by the Cre-induced *ihCD59* lines are comparable to those obtained with the widely used reporter Tdt

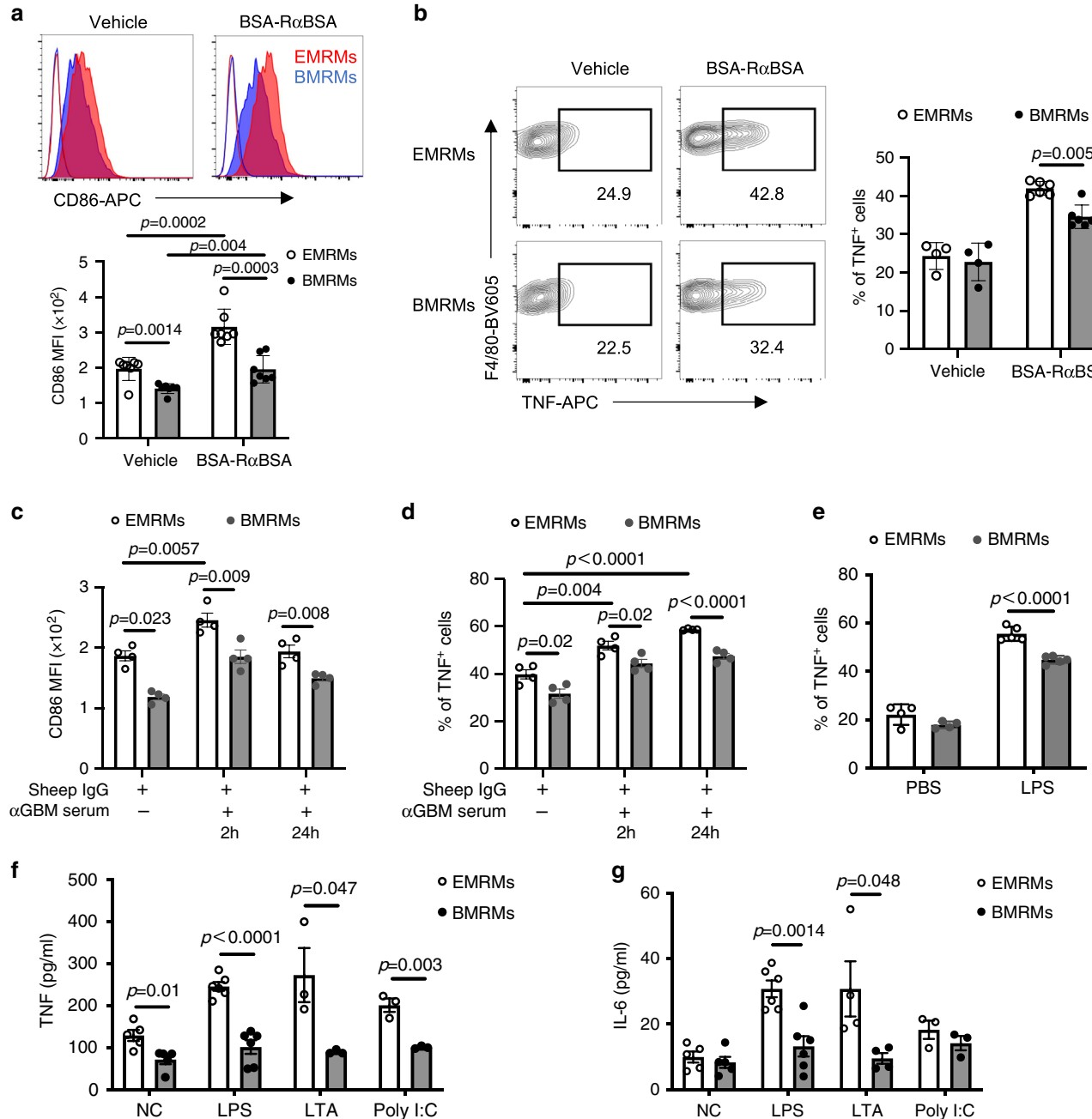

**Fig. 7 EMRMs have higher immune capacity than BMRMs.** EMRMs are more sensitive to immune challenges than BMRMs in vivo (**a**–**e**) and in vitro (**f**, **g**). **a** Representative histogram (upper panel) and MFI (lower panel) of CD86 expression on hCD59+ EMRMs and hCD59− BMRMs 2 h after PBS (Vehicle) or BSA-RαBSA injection. ($n = 7$ mice per group). **b** The percentage of TNF positive staining in EMRMs and BMRMs 2 h after PBS (Vehicle) or BSA-RαBSA injection. Left panel is representative dot plots. Right panel is the percentage of TNF+ cells ($n = 4$ mice for vehicle group; $n = 6$ mice for BSA-RαBSA group). **c** MFI of CD86 expression and **d** TNF staining in EMRMs and BMRMs 2 h and 24 h after αGBM serum injection in sheep IgG-preimmunized female mice. Data are pooled from two independent experiments ($n = 4$ mice per time point per group). **e** TNF staining in EMRMs and BMRMs 3 h after PBS or 5 μg/g b.w. LPS injection in male mice. Data are pooled from two independent experiments. ($n = 4$ mice for PBS group; $n = 5$ for LPS group). **f** TNF and **g** IL-6 level in the supernatant of sorted EMRMs and BMRMs treated with medium (NC), 100 ng/ml LPS, 5 μg/ml LTA, or 5 μg/ml Poly (I:C) for 18 h. Data are pooled from three independent experiments, each dot represents cells obtained from individual sorting pooled from two mice. ($n = 5$ for NC; $n = 6$ for LPS; $n = 3$ for LTA and Poly (I:C)). All data are presented as mean ± s.e.m. $p$-values by two-tailed unpaired $t$-test are indicated in **a**–**g**. Source data are provided as a Source Data file.

line[42]. Furthermore, Cre-induced *ihCD59* can be utilized to target a specific cell population, and, when coupled with ILY injection, to conditionally ablate the targeted cells to investigate the molecular mechanisms underlying cellular regeneration. Importantly, ILY-mediated cell death is very rapid (in seconds)[27,41]. This allows us to dynamically monitor the newly regenerated RMs and explore the niche signaling underlying regeneration/repopulation after the RMs

niche is emptied. Such studies can be challenging to perform with other widely used cell ablation models, including irradiation, clodronate, or DT/DTR[4,18–20,41]. ILY can easily access cell populations in tissues such as liver and kidney, but not in CNS[27]. Therefore, we were able to use ILY to rapidly ablate hCD59-labeled RMs. Taken together, these results demonstrate that the ILY/ihCD59-mediated rapid ablation method is very useful for simultaneously mapping

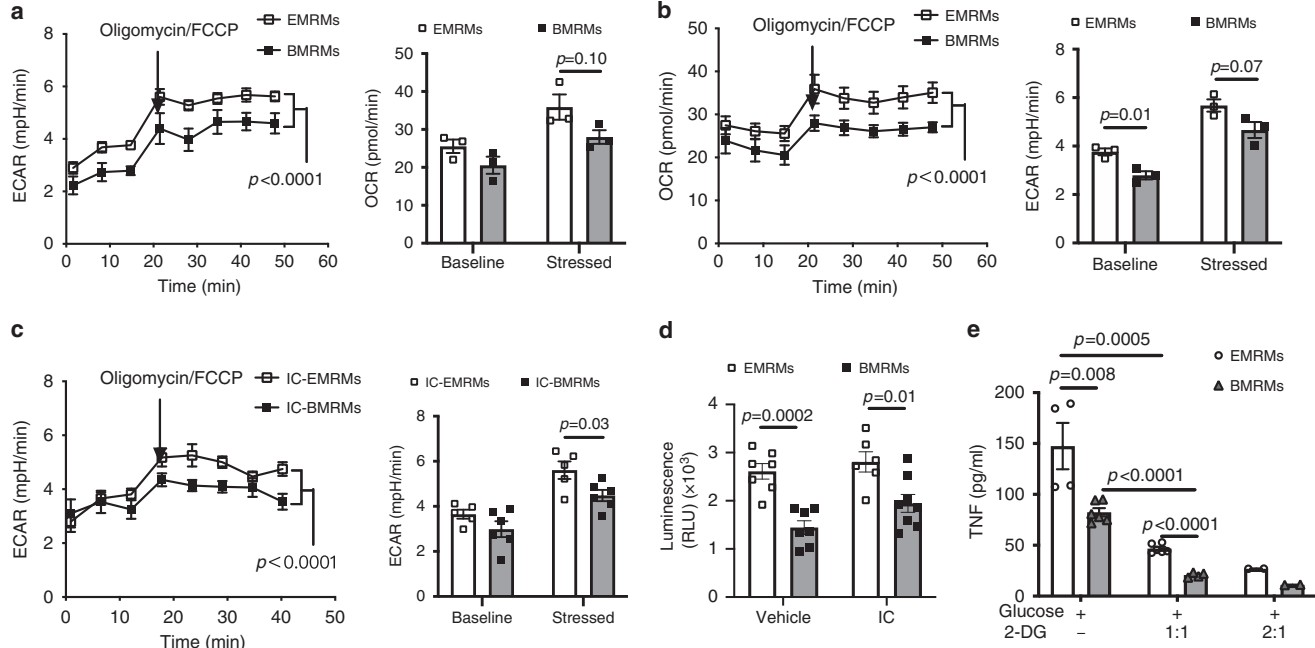

**Fig. 8 EMRMs have a greater glycolytic capacity than BMRMs. a, b** Real-time changes and quantifications of the ECAR (**a**) and OCR (**b**) in EMRMs and BMRMs at baseline or stressed with oligomycin/FCCP. Data are representative of three independent experiments ($n = 3$ per group). **c** Real-time changes and quantification of the ECAR in EMRMs and BMRMs sorted from Immune complex (IC)-treated mice at baseline or stressed with oligomycin/FCCP. Data are pooled from two independent experiments. ($n = 5$ for IC-EMRMs; $n = 6$ for IC-BMRMs). **d** Ex vivo glucose uptake in EMRMs and BMRMs sorted from PBS (Vehicle) or IC-treated mice. RMs were sorted and treated with 2-DG for 30 min, followed by measuring the level of its product 2DG6P by luminescence reader. Data are pooled from two independent experiments. (Vehicle: $n = 7$ for EMRMs and BMRMs; IC: $n = 6$ for EMRMs and $n = 8$ for BMRMs). **e** TNF production in LPS-stimulated EMRMs and BMRMs as in Fig. 7f together with or without glucose analog 2-DG. Data are pooled from two independent experiments and each dot represents cells obtained from individual sorting. (2-DG - group: $n = 4$ for EMRMs and $n = 6$ for BMRMs; 2-DG-1:1 group: $n = 6$ for EMRMs and $n = 4$ for BMRMs; 2-DG-2:1 group: $n = 2$ for EMRMs and BMRMs). All data are presented as mean ± s.e.m. $p$-values by two-way ANOVA analysis are indicated in the left panel of **a–c** and by two-tailed unpaired $t$-test in the right panel of **a–c** and **d**, **e**. Source data are provided as a Source Data file.

cell fate and studying niche signaling at least in tissues such as liver and kidney.

Our findings that postnatal expansion and maturation of RMs parallel renal growth indicates that RMs, as a structural unit of the kidney, may contribute to its development. This is in line with the current view that, in addition to their role as immune sentinels, TMs act as integral components of host tissue[62], orchestrate tissue remodeling, and contribute to the morphogenesis of developing tissues such as kidney[12,63]. In addition, using *Cx3cr1* promoter to conditionally express reporters for lineage tracing at embryonic and adult stages, we also document that RMs are primarily derived from fetal liver prenatally and that they self-maintain with a consistent contribution from peripheral monocytes throughout adulthood. After the RMs niche is emptied, peripheral monocytes rapidly enter the kidney and differentiate into RMs. The observation that reporters such as hCD59 and Tdt can label all RMs at a chosen time indicates that CX3CR1 is persistently and exclusively expressed in RMs, similar to microglia in the CNS[37]. This feature of CX3CR1 expression in RMs and microglia facilitates comprehensive and systematic investigation of their fate, origin, and dynamics under normal and pathological conditions[64]. The results of our studies using tamoxifen injection to label CX3CR1 + RMs during embryogenesis (E8.5,13.5, or 18.5) documented that prenatal RMs are primarily derived from fetal liver, not yolk sac, which is consistent with the findings obtained through lineage-tracing the progeny of labeled RMs progenitors[1,2]. Interestingly, the fetal liver derived RMs persist for at least 1 year through self-renewal and are supplemented by a rapid influx of peripheral monocytes after birth. Each of those

two pools contributes to ~50% of the total RMs, which eventually reaches a steady state. This ontogenic heterogeneity is distinct from other TMs studied so far[4]. TMs in brain, liver, lung, and skin, for example, are primarily derived during embryogenesis with, at most, only a minimal contribution from peripheral monocytes[4]. TMs in gut and heart also originally derive during embryonic stages but are eventually replaced completely by peripheral monocytes after birth in gut and at 2 months of age in heart[4]. TMs in the arterial wall arise from embryonic CX3CR1+ precursors and circulating monocytes immediately after birth but are maintained independently of circulating monocytes under physiological conditions throughout adulthood. Here, we found that at steady state, the pool of total RMs consists of a mixture of long-lived cells originating during embryogenesis and macrophages, which originate from BM postnatally and have a unique fate and dynamics.

Our results showed that RMs can be rapidly replenished from peripheral monocytes, almost half of the cells were recovered by day 7 after emptying the pool. Rapid replacement by either in situ proliferation of embryonic-derived cells, BM-derived monocytes, or both has been reported after ablating TMs in liver, brain, and lung[19,20,47]. Interestingly, RMs derived from peripheral monocytes after emptying the RMs niche are equally long-lived as the original, embryo-derived, RMs. This has also been reported in artery[32], heart[65], brain[66], and lung[67], but not liver[68]. Strikingly, circulating monocytes have a short half-life of a few days, as shown in Supplementary Fig. 2[22]. The dramatic differences between the half-life of BMRMs and monocytes clearly indicates that specific mechanisms or niche signals in kidney, dictate

differentiation, reprogramming, and maintenance of BM-derived RMs, which warrant further investigation[12,22].

Cytokines/chemokines, specific metabolites, local tissue environment, and transcription factors all contribute to organ-particular niche signals for TMs regeneration/repopulation[16]. Our results demonstrate that CX3CR1/CX3CL1 axis is required for embryonic development, migration, and continuous maintenance of RMs. The role of CX3CR1/CX3CL1 in the progression of various diseases in tissues, including kidney, has been recognized for many years[69]. However, its critical role in the niche signaling for RMs regeneration has not been studied before. Previous studies indicate that CX3CR1/CX3CL1 provides crucial niche signals in brain, arterial wall, and intestine. The number of microglial cells in brain of $Cx3cr1^{-/-}$ animals, for example, has recovered significantly slower than those of wild type at days 7 and 10, remains significantly lower at day 14, but reaches similar levels after day 32[70]. These data indicate that the CX3CR1/CX3CL1 axis potentiates microglial proliferation and morphological maturation but not maintenance. In the vascular niche, survival of resident arterial macrophages depends on the CX3CR1/CX3CL1 axis, but whether their maintenance also requires it is currently unknown. Additionally, the CX3CR1/CX3CL1 axis is critical for regulating migration and/or retention of a subset of intestinal macrophages that reside within the mucosal lamina propria (LP), but not for all TMs in the intestine[71]. Interestingly, the numbers of intestinal macrophages in CX3CR1-deficient mice are similar to wild-type[72], indicating that this axis may not be critical in the development of intestinal macrophages. In contrast, we found that at a steady state, deficiency of CX3CR1, but not of CCR2, significantly reduces the number of RMs from birth through adulthood but not microglia[24]. We also found that the CX3CR1/CX3CL1 axis is indispensable for specific regeneration and maintenance of RMs under pathologic conditions. These results indicate that the precise roles of this axis may differ for TMs in various tissues.

Emerging evidence indicates that the different subsets of TMs are functionally diverse[2]. It is widely accepted that the role of TMs differs from those of monocytes and infiltrating/surveilling macrophages. TMs, such as microglia and gut macrophages differ significantly between two tissues and even within different regions of the same tissue[64,73–75]. Various combinations of markers, including MHCII/CX3CR1/CD11b for macrophages in heart and F4/80 high/low for microglia, have been used to compare functional differences[64], but these markers may indicate activation status, rather than the origin, of the cells. Because the appropriate tools have been lacking, no studies have directly compared the intrinsic function and metabolic status of TMs of different origins, including embryo vs. BM-derived tissue macrophages. Here, for the first time, we have separated EMRMs from BMRMs by inducing expression of a reporter on EMRMs, but not on BMRMs, during embryogenesis to investigate the distinct functions of these two subpopulations of RMs. We studied the specific and relevant functions played by RMs: immune surveillance and clearance of immune complex, and immune response to major pathological substances released from bacteria and viruses in vivo and ex vivo. We demonstrate that EMRMs have a greater capacity for scavenging IC and are more sensitive to immune challenges than BMRMs. These results indicate that the origin of RMs significantly contributes to their function. In addition, we observed that EMRMs have higher expression of CX3CR1 and F4/80, and BMRMs have higher levels of CCR2 and CD11c. Differential expression of these classic myeloid cell markers, indicates these two RM populations possess intrinsic differences, which warrants further investigation. Interestingly, EMRMs have a higher metabolic capacity to produce energy than BMRMs and inhibiting glycolysis in RMs reduces the immune response. Recent in vitro studies have consistently shown that tissue-resident peritoneal macrophages have higher glycolytic capacity than BM-derived macrophages[26]. Our observation that inhibition of glycolysis in RMs reduces TNF release is also in accord with earlier studies[76–78]. One potential mechanism for reduced TNF release under conditions of low glycolysis is post-transcriptional repression as a result of interaction between glyceraldehyde 3-phosphate dehydrogenase (GAPDH) and *tnf* mRNA[79]. Furthermore, it is notable that some baseline values of two populations are also different, which may contribute to the differences in immune response and metabolic profile under the various challenges. Taken together, our results support the current notion that the function of macrophages in various contexts may be regulated by manipulating their metabolism[25,26] and also, for the first time, demonstrates that the origin of tissue-resident macrophages has a distinct purpose. Furthermore, we observed that EMRMs preferentially localize in medulla/pelvis. This suggests the role of anatomic or metabolic niches in shaping the unique feature of EMRMs, which may help to explain functional and metabolic differences between the two RMs populations as reported here. In addition, the heightened immune response of EMRMs may result from epigenetic modifications of metabolic signaling pathways at the chromatin level, which lead to functional modification or reprogramming of innate immune sensors and key inflammatory responses during embryonic development. Deciphering these mechanisms in future will improve understanding of the biology of RMs and allow selective modulation of immunity in autoimmune-related nephritis and acute kidney diseases.

## Methods

**Mice**. CD45.1 C57BL/6 (JAX 002014), CD45.2 C57BL/6 (JAX 000664), *Cx3cr1CreER*^+/+^ (JAX 021160), *Ccr2*^−/−^ (JAX 004999), *R26-Tdtomato*^+/+^ (JAX 007905) mice were purchased from The Jackson Laboratory (Bar Harbor, ME) and housed in Lewis Katz School of Medicine at Temple University. The *ihCD59*^+/+^ mice were previously generated and backcrossed with C57BL/6 background for at least seven generations[27]. All other mice used in this study were on C57BL/6 background. Homozygous *Cx3cr1CreER*^+/+^ mice are CX3CR1-deficient (*Cx3cr1 KO*) and heterozygous *Cx3cr1CreER*^+/−^ mice are CX3CR1-sufficient[36]. *Cx3cr1CreER*^+/+^ mice were crossed with *ihCD59*^+/−^ or *R26-Tdtomato*^+/−^ mice to generate *Cx3cr1CreER*^+/−^/*ihCD59*^−/−^, *Cx3cr1CreER*^+/−^/*ihCD59*^+/−^ or *Cx3cr1CreER*^+/−^/*R26Tdt*^−/−^, *Cx3cr1CreER*^+/−^/*R26Tdt*^+/−^ mice, respectively. *Cx3cr1CreER*^+/−^/*ihCD59*^+/−^ were further crossed with *Cx3cr1CreER*^+/+^ to generate CX3CR1-deficient *Cx3cr1CreER*^+/+^/*ihCD59*^+/−^ and CX3CR1 sufficient *CX3CR1CreER*^+/−^/*ihCD59*^+/−^ mice. Both male and female were used except the experiments in which the gender was specified. All animal experiments were reviewed and approved prior to commencement of the activity by the Animal Care and Use Committee from Tulane University (permit numbers 633 and 638) and Temple University (permit numbers 4598 and 4738). Mice were housed in the animal facility of Tulane University School of Medicine or Lewis Katz School of Medicine at Temple University, with a 12-h light/dark cycle, in specific-pathogen-free environment.

**Tamoxifen treatment**. Tamoxifen (Sigma-Aldrich) was dissolved in corn oil at 20 mg/ml. To induce Cre-mediated hCD59 or Tdt expression in the embryo, pregnant female was treated with 100 μg per g body weight (b.w.) of tamoxifen supplemented with 37.5 μg/g progesterone (Sigma-Aldrich) at 9, 14, or 19 days post coitum (E8.5, E13.5, or E18.5, respectively) through intraperitoneal (i.p.) injection. To induce Cre-mediated hCD59 or Tdt expression in adult, 8–12-week-old mice were treated with 100 μg/g tamoxifen for 3 consecutive days through i.p. injection.

**Immunostaining of renal macrophages**. In all, 8–12-week-old *Cx3cr1Cre*^+/−^ mice or E18.5-tamoxifen-labeled *Cx3cr1Cre*^+/−^/*R26Tdt*^+/−^ mice were euthanized by carbon dioxide ($CO_2$) and perfused with PBS. Kidneys were excised and fixed in 4% paraformaldehyde overnight. After washing with PBS, kidneys were immersed in 30% sucrose overnight, snap frozen in optimal cutting temperature (OCT) compound and sectioned (10 μm). Slides were immunostained with 1:400 dilution of rabbit anti-GFP antibody (A-11122, Invitrogen) overnight at 4 °C, and visualized with 1:500 dilution of Alexa 488-conjugated goat anti-rabbit IgG antibody (Invitrogen). Images were assessed and acquired at ×10 or ×20 magnification on a Leica DMRE Research Epi-Fluorescence microscope. In all, 2–4 views (×20) were randomly acquired from the cortex or medulla and the average cell counts were quantified by individual who is blinded to the experimental design.

**Intermedilysin purification**. His-tagged recombinant ILY was purified using His-band purification kit (EMD 70239) according to manufacturer's protocol[27,41]. ILY's concentration and purity were determined by SDS-PAGE.

**Renal macrophage depletion and regeneration**. Tamoxifen-treated *Cx3cr1CreER*[+/−], *Cx3cr1CreER*[+/−]*/ihCD59*[+/−], and *Cx3cr1CreER*[+/+]*/ihCD59*[+/−] mice were injected with one dose of ILY (120 ng/g b.w.) or PBS via tail vein (i.v.). In total, 24 h later, the ratio and number of RMs and microglia were determined by flow cytometry to confirm the depletion. RMs' regeneration was monitored at different time points afterward. In CCR2 antagonism experiment, we started to subcutaneously administrate CCR2 antagonist RS-504393 (Tocris) (2 mg/g, daily for 7 days) to the mice prior to the one dose of ILY injection.

**Bone marrow transplantation**. In total, 8-week-old male mice were lethally irradiated (9.5 Gy) and reconstituted with 10^6 donor BM cells by i.v. injection. The reconstitution rates in blood and tissue were determined by the ratio of the CD45.1 vs. CD45. 2 positive cells at 5 weeks after irradiation. For kidney protected BMT, mice were anesthetized by i.p. injection of ketamine (150 mg/kg) and xylazine (10 mg/kg) and positioned beneath a 5-mm-thick lead shield that covered the lower body including the kidney. Mice received partial irradiation (9.5 Gy) on exposed head, thorax, and forelimbs but not kidney. After recovery from anesthesia, mice were reconstituted with 10^6 donor BM cells by i.v. injection.

**Measurement of glomerular filtration rate**. As described in Qi et al.[48] with slight modifications. Briefly, male mice were injected with 5% FITC-inulin (Cayman) via tail vein at 3.74 µl/g body weight. 25 µl blood was collected using a heparinized capillary tube (VWR) from tail vein at 3, 7, 10, 15, 35, 55, and 75 min post injection. Plasma was used to determine the FITC concentration using BMG FLUOstar optima microplate reader with the excitation at 485 nm and emission at 538 nm. The concentration of FITC-inulin in plasma was calculated using the standard curve of serial FITC-inulin dilution. GFR is calculated using a two-compartment model of two-phase exponential decay (GraphPad Prism). The formula for GFR calculation is $GFR = I/(A/\alpha + B/\beta)$, where I is the amount of FITC-inulin delivered by the bolus injection, A and B are the Y-intercept values of the two decay rates, and $\alpha$ and $\beta$ are the decay constants for the distribution and elimination phases, respectively.

**Kidney ischemia-reperfusion injury**. Mice were anesthetized and kept on a homeothermic pad to maintain body temperature at 36.5 °C during the surgery. Midline incision was made to expose renal pedicles. Vascular clips were applied to the hilum of one (unilateral) or both (bilateral) kidneys to induce 30 min of kidney ischemia. After 30 min, the clips were released for reperfusion and then the abdomen was closed. Control mice were subjected to sham operation only without renal clamping. Serum and kidney were collected 18 h after surgery, then were used to analyze the serum chemokines and RMs, respectively.

**Anti-glomerular basement membrane nephritis model**. In total, 8–12-week-old male and female mice were pre-immunized intraperitoneally with 1 mg of normal sheep IgG (Jackson Immuno Research) in Freund's complete adjuvant (Sigma) and injected i.v. with 7 µl/g BW of sheep anti-rat GBM serum (Probetex) 7 days later. Control mice underwent same procedure without the injection of anti-GBM serum. Serum level of blood urea nitrogen (BUN) was measured at 1, 2, 5, and 7 days after anti-GBM serum injection using Urea Nitrogen Colorimetric detection kit (Thermo Scientific). Spot urine was collected before, 2 h and 24 h after anti-GBM serum injection. Urinary albumin and creatinine concentration were measured with mouse albumin ELISA kit (Bethyl Laboratories) and creatinine colorimetric assay kit (Caymen). The expression of CD86 and TNF in EMRMs and BMRMs were analyzed 2 h and 24 h after anti-GBM injection.

**Preparation of single cells from mouse tissue**. Mice were euthanized by CO_2 and perfused with PBS. The brain, lung, spleen, kidney, liver, heart, and femurs were harvested and processed for single cell preparation. BM single cells were flushed out from femurs using a syringe filled with cold PBS and kept on ice. Spleen and liver were mashed with plungers of 5 ml syringe in PBS and passed through 70 µm cell strainers and maintained on ice. Brain, lung, heart, and kidney were mechanically dissociated and incubated for 40 min in 10 ml 1xHBSS containing 0.75 mg/ml collagenase IV (Worthington) and 20 µg/ml DNaseI (Sigma) at 37 °C. After digestion, cell suspension was passed through 40 µm cell strainers and maintained on ice. The suspension was washed twice with PBS (500 × g, 8 min) before flow cytometry staining. To enrich hematopoietic cells, single cell suspensions from liver, brain, and kidney were further processed by Percoll gradient centrifugation. Briefly, the pellet was resuspended with 10 ml 37% Percoll and carefully layered over 3 ml 70% Percoll prior to density centrifugation (500 × g 30 min with no brake). After centrifugation, the layer at the junction of 37/70% Percoll was collected and washed twice with 10 ml PBS (500 × g, 8 min). Cell pellet was further washed in flow cytometry staining buffer (PBS with 2% BSA) and resuspended in 100 µl staining buffer and processed for staining.

**Flow cytometry staining, acquisition, and sorting**. Single cells suspension was incubated with 1:200 anti-CD16/32 (FcγRIII/II) for 15 min to block non-specific Fc receptor binding. The antibody cocktails were added and incubated at 4 °C in the dark for 30 min. The antibodies used are listed in Supplementary Table 1. In general, a 1:100 dilution was used. DAPI (Invitrogen) and SPHERO AccuCount Particles (Spherotech, Inc) were added before acquisition to distinguish live/dead cells and access absolute cell numbers, respectively. For intracellular Ki67 staining, cells were then fixed and permeabilized using the Foxp3 fix/perm buffer (eBioscience), followed by staining for 45 min with anti-Ki67-PE (or APC) or isotype control (eBioscience) in permeabilization buffer. For intracellular cytokine staining, cells were cultured 4 h at 37 °C in 5 mL DMEM supplemented with 10% FBS and 1:1000 Brefeldin A (Biolegend). After 4 h, cells were washed in staining buffer, blocked with purified anti-CD16/32 (FcγRIII/II) and stained for surface antigens. Then cells were fixed with IC fixation buffer (Biolegend) for 20 min, followed by permeabilization in permeabilization buffer (eBioscience) for 30 min. Cells were stained for 45 min with anti-TNF APC or PE (Biolegend) or isotype control (Biolegend) in permeabilization buffer (1:100). Data was acquired on a LSRII (BD) or LSRFortessa (BD) flow cytometer. All data were analyzed using FlowJo 10.6.1 (Tree Star, Inc.). RMs sorting was performed on Aria IIµ (BD) sorter. 1 mM EDTA was added to all buffers after kidney digestion. RMs were directly sorted into cold PBS containing 10% FBS and washed with DMEM complete medium containing 10% FBS after sorting. DAPI was used to exclude dead cells.

**In vivo uptake of immune complex**. Immune complex (IC) was prepared in vitro by incubating 50 µg BSA-Alexa647 (Invitrogen) with 110 µg rabbit anti-BSA IgG (RαBSA) (Invitrogen) at 4 °C for 1 h to generate Alexa647-BSA-RαBSA. IC was freshly generated and introduced by tail vein injection. IC uptake was determined by the density of Alexa647 fluorescence in RMs 2 h after injection.

**Ex vivo glucose uptake assay**. The uptake of glucose by RMs was measured using Glucose Uptake-Gl Assay kit (Promega). Briefly, 15,000 sorted EMRMs and BMRMs were washed extensively with PBS and incubated with 1 mM 2-deoxyglucose (2-DG) in PBS for 30 min at 37 °C. Following cell lysis, 2-deoxyglucose-6-phosphate (2DG6P), the metabolic product of 2-DG, was incubated with detection reagent for 1 h. Luminescence was measured by BMG FLUOstar optima microplate reader. Background luminescence was subtracted from the signals of all samples.

**In vivo and in vitro stimulation of renal macrophages**. Sorted EMRMs and BMRMs were incubated in DMEM containing 10% FBS for 4 h to recover from the sorting procedure. Afterwards, cells were treated with medium (control), 100 ng/ml LPS (O111:B4, Sigma-Aldrich), 5 µg/ml LTA, or 5 µg/ml Poly (I:C) (Invivogen) for 18 h. The supernatant was collected for cytokine measurement. For in vivo stimulation, we utilized an LPS-induced acute kidney injury model in male mice[80]. In total, 8–12-week-old male mice were treated with LPS (5 µg/g b.w.) (O111:B4, Sigma-Aldrich) through i.p. injection. RMs were isolated for CD86 and TNF staining 3 h after the injection.

**Cytokine/chemokine measurement**. Cytokines (IL-1α, IL-1β, IL-6, IL-10, IL-12p70, IL-17A, IL-23, IL-27, MCP-1, IFN-β, IFN-γ, TNF, and GM-CSF) and chemokines (CCL2, CCL5, CXCL10, CCL11, CCL17, CCL3, CCL4, CXCL9, CCL20, CXCL5, CXCL1, CXCL13, and CCL22) in the cell culture supernatant and serum were measured by LEGENDplex mouse proinflammatory chemokine panel and LEGENDplex mouse inflammation panel (Biolegend). CX3CL1 level was measured by mouse CX3CL1/Fractalkine Quantikine ELISA Kit (R&D).

**Seahorse metabolic analysis**. The cellular energy metabolism phenotypes of EMRMs and BMRMs were determined by Agilent Seahorse XFp or XFe24 Extracellular Flux Analyzer using the energy phenotype test kit (Agilent). This technique measures mitochondrial respiration and glycolysis under baseline and stressed conditions, to reveal two parameters of cell energy metabolism: baseline phenotype and stressed phenotype. The stressed metabolic phenotype of EMRMs and BMRM cells was determined using oligomycin (an inhibitor of ATP synthase), and FCCP (a mitochondrial uncoupling agent). A day before the assay, Agilent Seahorse XFp sensor cartridge was equilibrated in Agilent Seahorse XF calibrant at 37 °C in a non-CO_2 incubator overnight. EMRMs and BMRMs were washed in Seahorse XF RPMI medium, pH 7.4 (Cat No. 103576-100) containing 1 mM pyruvate, 2 mM glutamine, and 10 mM glucose. EMRMs and BMRMs (20,000 cells/well) were plated in triplicate in cell culture plates in a volume of 180 µl of RPMI medium, pH 7.4 containing 1 mM pyruvate, 2 mM glutamine, and 10 mM glucose for 1 h in a 37 °C non-CO_2 incubator. In all, 20 µl of stressor mix (10 µM of oligomycin and 10 µM of FCCP) was added into every port A of the hydrated sensor cartridge, which was then transferred to a temperature-controlled (37 °C) Seahorse Extracellular Flux Analyzer and subjected to an equilibration period. Following calibration of the sensor cartridge, the cell culture plate was then transferred to the Seahorse Extracellular Flux Analyzer and programmed for three baseline measurements and five measurements post-injection of the stressor. Upon completion of the assay the Agilent Seahorse XF Cell Energy phenotype test report generator was used for analysis of the data.

**Statistics**. Data are presented as mean ± s.e.m. To compare values obtained from multiple groups over time, two-way analysis of variance (ANOVA) was used, followed by Bonferroni post hoc test. To compare values obtained from two groups, two-tailed unpaired Student's $t$-test was performed. Statistical significance was taken at the $p < 0.05$ level, n.s. indicates $p > 0.05$.

**Reporting summary**. Further information on research design is available in the Nature Research Reporting Summary linked to this article.

## Data availability

The source data underlying Figs. 1b–d, 2b–f, 3b, d, 4b, c, e, g, h, 5a–c, e, f, 6c–f, 7a–g, and 8a–e and Supplementary Figs. 3a, b, 5b, c, 6a, b, 7a, b, 8a–d, 10–13 are provided as a Source Data file. All other data are available from the corresponding authors upon reasonable requests.

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

## Acknowledgements

This work was supported by R21OD024931 (X.Q.), R01 HL130233 (X.Q.), R01HL141132 (X.Q.), NS094834 (P.V.G.K.), PO1MH105303 (T.F.), P20GM103629-08S1 (T.F.), 5P01DA037830-05 (P.K.D.), and 5 P51OD011104-58 (J.R.).

## Author contributions

F.L., S.D., D.F., Z.Q., X.P., S.S.V.P.S., M.R., L.H., M.C., K.E.M., P.Q., Y.C., C.Z., F.Z., S.L., B.H.A., X.Y., H.W., P.V.G.K., D.W.B., T.F., P.K.D., J.R., B.G., and X.Q. designed and performed experiments. F.L., S.D., D.F., Z.Q., X.P., S.S.V.P.S., L.H., M.C., K.E.M., P.Q., Y.C., C.Z., F.Z., S.L., T.F., P.K.D., and X.Q. analyzed data. F.L., B.H.A., X.Y., H.W., P.K.D., T.F., J.R., B.G., and X.Q. wrote the first draft of the manuscript and all authors participated in the review and critique of the manuscript. F.L. and X.Q. supervised the project, designed experiments, analyzed data, and finalized the writing of the paper.

## Competing interests

The authors declare no competing interests.
