## [Peer Review File · Nature Communications]

Reviewers' comments:

Reviewer #1 (Innate immunity, immune metabolism)(Remarks to the Author):

This is an interesting and timely assessment of the origin, turnover, and some of the functional characteristics of renal macrophages (RM). The paper uses a variety of creative genetic models to assess origin and turnover. I found the fidelity of the system to be high, and the assessment included well thought out controls that support the use of a novel system involving hCD59 transgenes and depletion of cells with ILY. The system seems valid and the use of the immediate deletion with ILY is nice.

Similarly, I found the elucidation of inflammatory changes in RM depending on their origin, although superficial, convincing.

Major Concerns:

-The functional characterization is superficial including only Ig uptake and cytokine production. More convincing models of disease would strengthen the paper. Can it be demonstrated that disease elicited by Ig deposition progresses differently when RM are of different origins?

-The metabolic assessment is superficial and appears to have been added as an afterthought. The changes shown, especially the relative lack of difference in the OCR/ECAR ratios suggests these cells are not substantially different metabolically. Assessment during stimulation with Ig is needed to draw conclusions, as is better mechanistic dissection. For example, are there differences in glucose uptake in vivo during Ig challenge or during reconstitution after ischemia? Does metabolic profiling support the Seahorse data? Finally, metabolic assessment in conditions associated with the renal environment would be more enlightening. Perhaps the EMRM are better adapted to the metabolic niche than the BMRM?

Reviewer #2 (Kidney disease, lupus nephritis)(Remarks to the Author):

These studies contain an exhausting number of various transgenic animals with varying time courses and experiments, to assess the time course and relative abundance of tissue resident macrophages, focusing on the kidney, comparing bone marrow-derived versus embryo-derived macrophage populations. Although the studies are quite extensive and shed light on some aspects of macrophage resident cells and their development and repopulation after depletion, and some in vivo manipulations are included, these do not include direct manipulation of the populations to examine any change of the phenotype of any resulting organ injury, or even the impact of the populations on development. The ingestion of injected immune complexes by either of the two macrophage populations is of interest, but there are no corollary studies of localization within the tissue or effects on tissue injury of this immune complex degradation.

Similarly, the inflammatory cytokine profiles from the populations are done in vitro without corollary in vivo experiments.

The n of animals in many studies is quite small- 4-5- were both male and female mice included? If so, was there any difference in responses according to sex?

In particular, the addition of any functional studies would be important to show that the radiation for the chimeric mice does not induce any subtle injury, and likewise, in aging, it would add to show any functional data, including albuminuria and impact on GFR of the various manipulations. Lastly,

localization of the cells by immunostaining of tissue sections would also help to provide a more relevant context for the extensive data, which is largely based on flow studies.

Comments from Reviewer #1

Comment 1: *The functional characterization is superficial including only Ig uptake and cytokine production. More convincing models of disease would strengthen the paper. Can it be demonstrated that disease elicited by Ig deposition progresses differently when RM are of different origins?*

Response: To address this specific comment, we performed the additional experiments and have presented the experimental results in Fig. 7. We investigated whether EMRMs and BMRMs differ in their response to immune challenge in the disease settings. Renal macrophages act as front line defenders against immune challenge and have been shown to scavenge immune complexes, and respond to anti-glomerular basement membrane (GBM) glomerulonephritis, and lipopolysaccharide (LPS)-induced renal inflammation. Utilizing CD86 and TNF- α as indicators of macrophage activation, we monitored the response of EMRMs and BMRMs to these challenges in vivo. Consistent with greater uptake of ICs (Fig. 6B-D), EMRMs expressed significantly higher levels of CD86 and TNF- α than BMRMs 2 hours after IC injection (Fig. 7A and 7B). In response to anti-GBM serum, EMRMs had increased CD86 expression and TNF- α production than BMRMs at 2 and 24 hours after challenge (Fig. 7C and 7D). This response by RMs occurred well in advance of evident renal injury, as indicated by increased serum BUN, which was not observed until day 2 after challenge (Fig. S8). Similarly, LPS challenge induced higher level of TNF- α in EMRMs than that in BMRMs in vivo (Fig. 7E). Although LPS challenge increased CD86 expression, it did not result in any significant difference between two populations (data not shown). Together, these results indicate that EMRMs are more sensitive to immune challenge than BMRMs. We have included these in the result section.

Comment 2: *The metabolic assessment is superficial and appears to have been added as an afterthought. The changes shown, especially the relative lack of difference in the OCR/ECAR ratios suggests these cells are not substantially different metabolically. Assessment during stimulation with Ig is needed to draw conclusions, as is better mechanistic dissection.*

Response: To address this specific comment, we further explored the different metabolic profiles of EMRMs and BMRMs in the pathological condition such as IC challenge. Total RMs from IC-injected mice had more ECAR than RMs from control animals (Fig. S10). This also supports previous observation of activated macrophages challenged with LPS. These results indicate that macrophages can shift energy metabolism from mitochondrial respiration to glycolysis under pathological conditions, which makes them resilient to survive low oxygen microenvironments. We consistently observed that EMRMs have higher ECAR than BMRMs after IC stimulation (Fig. 8C).

Comment 3: *For example, are there differences in glucose uptake in vivo during Ig challenge or during reconstitution after ischemia? Does metabolic profiling support the seahorse data?*

Response: In our additional glucose uptake experiment, we consistently observed that EMRMs have a higher glucose uptake than BMRMs under static and stimulating conditions *ex vivo* (Fig. 8D), which ties metabolic activation of EMRMs and BMRMs with their glucose uptake.

Comment 4: *Finally, metabolic assessment in conditions associated with the renal environment*

would be more enlightening. Perhaps the EMRM are better adapted to the metabolic niche than the BMRM?

Response: Thanks for the great comment. It is conceivable that the EMRMs are better adapted to the metabolic niche than the BMRMs, which explain these functional and metabolic differences as reported here. However, the mechanisms underlying these intrinsic functional and metabolic differences between these two populations of RMs are unknown and warrant further investigation. We have discussed this issue in discussion as following: “It is conceivable that the EMRMs are better adapted to the metabolic niche than BMRMs, which may help explain functional and metabolic differences between the two RM populations as reported here. For example, the heightened immune response of EMRMs may result from epigenetic modifications of metabolic signaling pathways at the chromatin level, which lead to functional modification or reprogramming of innate immune sensors and key inflammatory responses during embryonic development. Deciphering these mechanisms in future will improve understanding of the biology of RMs and allow selective modulation of immunity in autoimmune-related nephritis and acute kidney diseases.”

Comments from Reviewer #2

Comment 1: *These do not include direct manipulation of the populations to examine any change of the phenotype of any resulting organ injury, or even the impact of the populations on development.*

Response: Thanks for the important comments. Extensive evidence indicates that renal macrophages (RMs) participate in tissue homeostasis, inflammation, immune complex clearance, and repair. RMs consist of embryo-derived (EMRMs) and bone marrow-derived RMs (BMRMs), but the fate, dynamics, replenishment, functions and metabolic states of the two RM populations remain unclear. Our current studies focus on characterization of renal macrophage (RM) fate, dynamics, and niches. We also document that embryonic RMs are more sensitive to immune challenges than bone marrow-derived RMs, which is correlated with their respective metabolic profiles. The impact of direct manipulation of the populations on organ injury, or on development is an important and interesting question, which warrants further extensive investigation in different experimental settings.

Comment 2: *The ingestion of injected immune complexes by either of the two macrophages populations is of interest, but there are no corollary studies of localization within the tissue or effects on tissue injury of this immune complex degradation.*

Response: To address this issue, we performed immunostaining studies in adult E18.5-Tam labeled *Cx3cr1CreER^{+/+}/R26Tdt^{+/+}* mice. As described in Fig. 3, we have found that EMRMs and BMRMs are located in medulla and cortex and are predominantly associated with renal tubules (Fig. 3).

Comment 3: *Similarly, the inflammatory cytokine profiles from the populations are done in vitro without corollary in vivo experiments.*

Response: To address this issue, we performed the LPS, IC and anti-GBM experiment in vivo. As described in Fig. 7, we investigated whether EMRMs and BMRMs differ in their response to immune challenge in the disease settings. Renal macrophages act as front line defenders against immune challenge and have been shown to scavenge immune complexes, and respond to anti-glomerular basement membrane (GBM) glomerulonephritis, and lipopolysaccharide

(LPS)-induced renal inflammation. Utilizing CD86 and TNF- α as indicators of macrophage activation, we monitored the response of EMRMs and BMRMs to these challenges in vivo. Consistent with greater uptake of ICs (Fig. 6B-D), EMRMs expressed significantly higher levels of CD86 and TNF- α than BMRMs 2 hours after IC injection (Fig. 7A and 7B). In response to anti-GBM serum, EMRMs had increased CD86 expression and TNF- α production than BMRMs at 2 and 24 hours after challenge (Fig. 7C and 7D). This response by RMs occurred well in advance of evident renal injury, as indicated by increased serum BUN, which was not observed until day 2 after challenge (Fig. S8). Similarly, LPS challenge induced higher level of TNF- α in EMRMs than that in BMRMs in vivo (Fig. 7E). Although LPS challenge increased CD86 expression, it did not result in any significant difference between two populations (data not shown). Together, these results indicate that EMRMs are more sensitive to immune challenge than BMRMs. We have included these in the result section.

Comment 4: *The n of animals in many studies is quite small- 4-5- were both male and female mice included? If so, was there any difference in responses according to sex?.*

Response: We specified the gender in the methods and figure legends, otherwise it's a mixture of male and female experiment, in which we did not observe a significant difference in the parameters tested between males and females.

Comment 5: *In particular, the addition of any functional studies would be important to show that the radiation for the chimeric mice does not induce any subtle injury, and likewise, in aging, it would add to show any functional data, including albuminuria and impact on GFR of the various manipulations.*

Response: To address this concern, we have added the experimental results described in Fig. S5. We demonstrated that the radiation and aging did not induce significant change in the glomerular filtration rate (GFR), a sensitive clinical biomarker for kidney function as compared with the respective control mice (Fig. S5).

Comment 6: *Lastly, localization of the cells by immunostaining of tissue sections would also help to provide a more relevant context for the extensive data, which is largely based on flow studies.*

Response: We provide additional results in Fig. 3 to address this concern.

Reviewers' comments:

Reviewer #1 (Remarks to the Author):

My previous concerns are largely addressed. The addition of the in vivo IC injections do in fact show differences between bone marrow derived renal macs. In addition, the additional metabolic assessment is a good addition, although still limited. It must be noted however that these data show small differences that seem to reflect the baseline values. This suggests that the response to stimulus is similar, but that because the baseline is different the final levels achieved are different. For example see 7A, C, D, and E. This obvious trend should be noted and acknowledged in the results and discussion. A similar phenomenon is clear in 8C and 8D (the metabolic data)

Reviewer #2 (Remarks to the Author):

The authors have provided some additional in vivo data. Point 1, "the impact of the populations on development/injury" has not been addressed other than to say "beyond the scope". Key elements of the added in vivo antiGBM model are missing. Although they show BUN data, there is no data on the time course of tissue injury. At 2 and 24 hrs, when they have measured CD86 and TNF alpha in the EMRM and BMRM, what is the tissue injury? Is there any glomerular necrosis or detectable injury? Is there hematuria or proteinuria at that time point? BUN is not the most sensitive marker for early injury. Adding the time course of their chosen model, and showing localization of the macrophage populations at the time points where they have measured functional markers of the cells would put a more relevant context on the data.

The response regarding sex differences is not scientifically rigorous- with only up to 5 mice in groups with a mix of male and female it is not possible to meaningfully statistically assess sex differences. It is also not clear why LPS is done in all male, and antiGBM in all female. These issues should at least be acknowledged within the manuscript, as many inflammatory, autoimmune and immune complex-related diseases have distinct differences clinically in males vs females, and the current studies do not allow any insights into potential differences in macrophage populations to explain these differences.

The localization studies shown in Fig 3 are not detailed sufficiently to understand relative abundance in cortex vs medulla; the authors state only the cells were mostly "associated with tubules". Since the kidney is >90% tubules, this is not particularly illuminating. They should provide a bit more granularity on the tissue localization.

Comments from Reviewer #1

Reviewer #1 (Remarks to the Author):

Comment: My previous concerns are largely addressed. The addition of the in vivo IC injections do in fact show differences between bone marrow derived renal macs. In addition, the additional metabolic assessment is a good addition, although still limited. It must be noted however that these data show small differences that seem to reflect the baseline values. This suggests that the response to stimulus is similar, but that because the baseline is different the final levels achieved are different. For example, see 7A, C, D, and E. This obvious trend should be noted and acknowledged in the results and discussion. A similar phenomenon is clear in 8C and 8D (the metabolic data)

Response: Based on the suggestion, we add the following sentences in the discussion:
“Furthermore, it is notable that some baseline values of two populations are also different, which may contribute to the differences in immune response and metabolic profile under the various challenges.”

Comments from Reviewer #2

Comment 1: Key elements of the added in vivo anti GBM model are missing. Although they show BUN data, there is no data on the time course of tissue injury. At 2 and 24 hrs, when they have measured CD86 and TNF alpha in the EMRM and BMRM, what is the tissue injury? Is there any glomerular necrosis or detectable injury? Is there hematuria or proteinuria at that time point? BUN is not the most sensitive marker for early injury. Adding the time course of their chosen model, and showing localization of the macrophage populations at the time points where they have measured functional markers of the cells would put a more relevant context in the data.

Response: We performed urine albumin-to-creatinine (ACR) analysis and IF staining of RMs after anti-GBM serum injection. We demonstrated that the response by RMs occurred in parallel with the onset of disease, as indicated by increased urine albumin-to-creatinine ratio (ACR) at 2h after anti-GBM serum injection. (Fig. S9A). Urine ACR is the early evidence of renal dysfunction and happened before the elevation of serum blood urine nitrogen, which was not

observed until day 2 after challenge (Fig. S9B). We did not observe a significant change of the localization of RMs before and 24h after anti-GBM serum injection. (Fig. S9C and S9D).

Comment 2: The response regarding sex differences is not scientifically rigorous- with only up to 5 mice in groups with a mix of male and female it is not possible to meaningfully statistically assess sex differences. It is also not clear why LPS is done in all male, and antiGBM in all female. These issues should at least be acknowledged within the manuscript, as many inflammatory, autoimmune and immune complex-related diseases have distinct differences clinically in males vs females, and the current studies do not allow any insights into potential differences in macrophage populations to explain these differences.

Response: We compared the response to GBM injection between male and female by determining the urine ACR and activation of RMs using CD86 expression. We have included the following data in the result section: "In addition, since many autoimmune and kidney diseases have gender differences in prevalence and clinical features, we compared the response of RMs to anti-GBM induced nephritis in male and female mice. We found a delayed disease onset in male mice, indicated by our observation that males had significantly lower urine ACR than females at 2h after anti-GBM serum injection. (Fig. S10A). Consistently, males had significantly lower CD86 levels on EMRMs and BMRMs than females at 2h after challenge (Fig. S10B). Also, we specified the utilization of male mice in LPS in method as following: "For in vivo stimulation, we utilized an LPS-induced acute kidney injury model in male mice as previously described [38]."

Comment 3: The localization studies shown in Fig 3 are not detailed sufficiently to understand relative abundance in cortex vs medulla; the authors state only the cells were mostly "associated with tubules". Since the kidney is >90% tubules, this is not particularly illuminating. They should provide a bit more granularity on the tissue localization.

Response: As suggested, we performed IF staining to quantify the localization of RMs in cortex vs medulla from *Cx3cr1CreER^{+/+}* mice. We also quantified the distribution of EMRMs and BMRMs using E18.5-Tam-*Cx3cr1CreER^{+/+}/R26Tdt^{+/+}* mice. Anatomically, immunofluorescent (IF) staining of Cx3Cr1-EYFP in kidney sections from *Cx3cr1CreER^{+/+}* mice showed that CX3CR1+ RMs spread throughout the cortex and medulla/pelvis, which is consistent with previous study (Fig. S8A). Interestingly, we observed that the density of EYFP+ RMs in medulla/pelvis was significantly higher than that of cortex (Fig. 3A and 3B). Furthermore, kidney sections from

E18.5-Tam-Cx3cr1CreER^{+/+}/R26Tdt^{+/+} mice revealed that EMRMs is preferentially located in medulla/pelvis, as evidenced by the significant higher percentage of Tdt+ labelling in medulla/pelvis than that in cortex. (Fig. 3C and 3D)

REVIEWERS' COMMENTS:

Reviewer #2 (Remarks to the Author):

The additional data have somewhat addressed the. previous comments.